# Narcissistic Chief Executive Officers and Their Effects on R&D Investment and Firm Performance: The Moderating Role of Managerial Discretion

**DOI:** 10.3390/bs14111115

**Published:** 2024-11-20

**Authors:** Qingzhu Gao, Liangmou Gao, Guangyan Zhang

**Affiliations:** 1School of Business Administration, Dongbei University of Finance and Economics, Dalian 116025, China; qingzhugao@126.com (Q.G.); liangmou@dufe.edu.cn (L.G.); 2Department of Management, Kyrgyz State University Named After I. Arabaev, Bishkek 720026, Kyrgyzstan; 3Department of Doctoral Studies, Kyrgyz State University Named After I. Arabaev, Bishkek 720026, Kyrgyzstan

**Keywords:** CEO personality, CEO narcissism, R&D investment, firm performance, managerial discretion

## Abstract

The impact of the chief executive officer (CEO) narcissism on a firm’s performance has gained attention from the academic community. However, the extant literature has largely ignored the mediating mechanism of research and development (R&D) investment and the moderating roles of managerial discretion. Additionally, the measurement of CEO narcissism is rarely disclosed in the public database. Compiling a CEO narcissism index from a video survey, we systematically explore the effect of CEO narcissism on firm performance, the mediating role of R&D investment, and the moderating role of managerial discretion. Based on the upper echelons theory, using a sample of 183 Chinese A-share listed manufacturing firms from 2011 to 2019, we found that CEO narcissism positively and significantly impacts R&D investment and firm performance, and then R&D investment mediated the relationships between CEO narcissism and firm performance. In addition, we found that managerial discretion could affect the relationship between CEO narcissism and R&D investment. Specifically, CEO duality and CEO ownership will strengthen the positive influence of a CEO’s narcissism in corporate R&D investment. Our results suggest that CEO narcissism appears to be a stimulus to corporate R&D investment; thus, in recruiting top executives, their psychological traits, especially narcissism, should be given special consideration.

## 1. Introduction

Firm value is essential for a company’s survival and growth. The upper echelons theory posits that CEO characteristics influence future firm performance by shaping firms’ strategic actions [1]. Prior research has shown that CEO demographic characteristics (e.g., CEO tenure, CEO duality, CEO nationality, CEO compensation, CEO experience) significantly impact firm performance [1,2,3,4]. These findings provide valuable insights into the relationship between CEO characteristics and firm performance. Moreover, the leadership styles and psychological traits of CEOs can profoundly influence a firm’s performance. Psychological research indicates that CEOs do not solely adhere to rational decision-making models, rather, their choices are profoundly shaped by their psychological traits [5,6]. Narcissism is an important component of a CEO’s psychological profile, reflecting their cognitions and values. Also, CEO narcissism is characterized by grandiosity, self-focus, and self-importance [7,8]. Recent evidence suggests that highly narcissistic CEOs tend to exaggerate their abilities, prioritize self-serving objectives, crave attention, and be motivated by self-interest and power. These tendencies will motivate narcissistic CEOs to engage in various activities, which may, in turn, influence firm performance [9,10].

Based on the upper echelons theory, scholars have explored the relationship between CEO narcissism and firm performance. However, the empirical literature yields mixed results. Scholars found that CEO narcissism could significantly increase firm performance. Highly narcissistic CEOs often assess risks and opportunities with heightened optimism and assertiveness [9], which will result in firm-beneficial outcomes [10]. Conversely, other scholars found that CEO narcissism is negatively associated with firm performance. Highly narcissistic CEOs frequently engage in overinvestment to attract public attention, which can lead to inefficiencies in investment, ultimately compromising organizational performance [8,11]. Some scholars also observed that the impact of CEO narcissism on firm performance remains uncertain [12]. Furthermore, the influence of CEO narcissism on R&D investment has become a focal point of interest. Several recent studies found that narcissistic CEOs crave attention and praise, tend to be dominant, and their desire for status and power propels them to take risks and pursue R&D activities [13]. R&D investment presents a substantial opportunity to attract media attention and stakeholder reactions, which, in turn, would satisfy the CEO’s personal needs for attention, and praise [14]. However, other research presents a contrasting view, indicating a negative correlation between CEO narcissism and green innovation [15]. Narcissistic CEOs may prioritize aggressive growth strategies aimed at building a “business empire” [16], which can ultimately lead to a decrease in R&D spending. Moreover, some recent findings suggest that there may be no significant relationship between CEO narcissism and R&D investment [17].

These inconsistent and contradictory effects may stem from several factors, detailed as follows: (1) Numerous studies have utilized survey data from various cultural backgrounds, highlighting how national cultural values can influence the personality traits of CEOs. For instance, narcissism is a trait often associated with Western cultures, whereas Chinese society is deeply rooted in Confucian principles and exhibits a largely collectivistic mindset. Consequently, CEOs from diverse cultural backgrounds possess distinct life experiences and values, which can lead to variations in their decision-making performance and work behaviors. (2) Prior research has explored the mediating effects of corporate social responsibility [18], earnings management [19], and ownership structure [6] on the relationship between CEO narcissism and firm performance, but has largely ignored the role of R&D investment. Narcissistic CEOs strongly desire to hold the spotlight by attracting appreciation and praise from an external audience [20]. This psychological trait has been used to underscore the link between CEO narcissism and a firm’s R&D investment decisions. The need for public attention drives narcissistic CEOs to pursue R&D investments, showcasing their vision, authority, and leadership. Therefore, it is essential to assess the effect of R&D investment. (3) Previous studies have neglected several key moderating factors that significantly influence the behavior of narcissistic CEOs. Narcissistic CEOs may encounter resource constraints and governance limitations that influence their decision-making processes, ultimately affecting R&D investment and firm performance. Cragun et al. [7] emphasize the importance of examining the factors that help narcissistic CEOs enforce their preference, underlining managerial discretion as an important factor.

The upper echelons theory posits that managerial discretion is a pivotal factor that determines the range of actions available to CEOs [21]. Managerial discretion refers to the latitude of managerial action [22]. Higher managerial discretion enhances CEOs’ confidence in their capacity to implement strategic initiatives [23]. In contrast, lower levels of CEO managerial discretion may enjoy less autonomy in decision-making. The effect of managerial discretion on narcissistic CEOs’ strategic behavior and decisions has received researchers’ attention. Multiple studies have found that the high managerial discretion influences organizational outcomes (business strategy, behavior, and performance) and increases the potential marginal benefits for CEOs (CEO total compensation and CEO power) [1,4]. Also, it moderated the relationship between CEO narcissism and behavioral decision-making (internationalization decisions, corporate social responsibility activities, earnings management behaviors, corporate acquisitions, and R&D investment). However, less attention has been paid to the moderate effect of CEO duality and ownership. CEO duality and ownership, reflecting high CEO power, would enhance the CEO’s managerial discretion [3]. Therefore, we hypothesize that these internal organizational factors such as CEO duality and ownership will influence the relationship between CEO narcissism and corporate decision-making.

Therefore, in this study, we apply the upper echelons theory to examine the influence of CEO narcissism on R&D investment and firm performance while also evaluating the mediating effect of R&D investment. Additionally, we investigate the moderating effect of managerial discretion (specifically, CEO duality and CEO ownership) on the relationship between CEO narcissism and R&D investment. Our contributions can be summarized as follows. First, our research enhances the understanding of CEO narcissism, R&D investment, and firm performance within the Chinese context. While the primary impacts of CEO narcissism on R&D investment and firm performance have garnered attention, much of the existing literature is mostly from developed country settings. We address this gap by examining the topic through the lens of Confucian values and ideologies, shedding light on how CEO narcissism influences R&D investment and firm performance. Notably, we confirm that R&D investment mediates the positive relationship between CEO narcissism and firm performance, providing insight into the “internal black box” that elucidates how CEO narcissism affects firm performance. Second, this study investigates the boundary conditions of the effect of CEO narcissism on R&D investment by introducing managerial discretion. We hypothesize that managerial discretion (CEO duality and CEO ownership) may strengthen the relationship between CEO narcissism and R&D. Empirical results support this hypothesis, thereby clarifying and reinforcing the relationship between CEO narcissism and R&D investment while enriching our understanding of the contextual factors influencing R&D investment.

## 2. Literature Review and Hypothesis Development

### 2.1. Upper Echelons Theory and Implications of CEO Narcissism on Firm Performance

The upper echelons theory posits that CEO characteristics significantly influence a firm’s future performance through strategic actions [24]. Innovation plays a crucial role in driving business growth, creating value, and establishing sustainable competitive advantages for enterprises [25]. CEO characteristics emerge as determinants of both innovation and business performance. Previous studies have investigated the potential relationships between CEOs’ personal traits, innovative behaviors, and firm performance [26]. CEO characteristics, including their leadership styles [27], and personality traits [28], can profoundly significantly influence a firm’s innovation strategies and performance. Some scholars have found that a CEO’s leadership style can affect a company’s innovation and performance. For example, transformational leadership positively influences firm performance, and business model innovation as a mediator of this relationship [29]. Proactive CEOs can shape exploitative innovation and, to some extent, exploratory innovation, and its effect on firm performance is partially mediated by organizational ambidexterity [30]. Servant leadership positively impacts the performance of third-sector organizations, with innovative capacity as a mediating factor in this relationship [31]. Ethical leadership enhances environmental performance through green IT capital and technology innovation [32]. Entrepreneurs’ extraversion, conscientiousness, and openness positively affect innovativeness, and indirectly positively affect technological improvement and new technology adoption [33]. Some scholars have found that the personality traits of CEOs can significantly impact corporate innovation and performance. For example, overconfident CEOs may amplify the positive effects of innovation premium on firm performance [34]. CEO humility promotes both exploitative and explorative innovation [35]. Executive hubris is positively correlated with firm innovation; however, this relationship diminishes in more complex environments [36]. Machiavellian CEOs can strengthen the relationship between strategic entrepreneurial behaviors and firm performance [37]. This literature deepens our understanding of the connections between CEO personal characteristics, innovative behavior, and firm performance. However, increasing scholars emphasize that studies should focus on the influence of CEOs’ dark personality traits. Shirokova et al. [37] suggest that future research should concentrate on the CEOs’ dark traits and leadership because this could enhance our understanding of leadership’s role in complex and dynamic contexts. Narcissism is often considered an important component of dark personality traits. Generally, CEOs are more likely to exhibit narcissistic personality traits.

Recent developments in the upper echelons theory propose that narcissism is a distinct psychological bias of CEOs [19]. Narcissistic CEOs often prioritize themes of power and self-centered objectives, driven by a strong desire for attention and praise [38]. Narcissism and overconfidence are closely related and very similar, but important differences exist. First, narcissism is regarded as a stable personality trait, whereas overconfidence tends to be more context-dependent [39]. Second, overconfident individuals tend to exhibit a sense of superiority without continuously seeking feedback from external sources to affirm their ego. In contrast, narcissists require constant validation from others. Evidence indicates that narcissistic individuals possess specific knowledge, skills, abilities, and personality traits that help them become managers [40]. Narcissistic CEOs exhibit both cognitive and motivational characteristics [20]. Cognitively, narcissistic CEOs maintain an inflated sense of self-worth. They are self-centered, confident in their abilities, and often perceive themselves as superior. Motivationally, narcissistic CEOs constantly seek affirmation, praise, and applause from others to create and maintain a sufficient “narcissistic supply”. The influence of CEO narcissism on organizational outcomes has attracted significant attention from scholars. This phenomenon has been particularly noted in the context of strategic decision-making processes. Research indicates that narcissistic CEOs influence the firms’ strategic choices and outcomes. Narcissistic CEOs have a penchant for engaging in risky initiatives, because they need constant and complete attention to maintain a positive and inflated self-view [41]. Several studies have reported that narcissistic CEOs tend to engage in exploitation and acquisition [38] and risky R&D projects [42]. The relationship between CEO narcissism and financial accounting has also been investigated [19]. Additionally, the accounting literature has focused on how narcissistic CEOs use their authority to impact the accounting mechanisms and standards to facilitate attention. For example, narcissistic CEOs tend to inflate aspects of their performance that are exposed to the media and public, such as CSR performance and earnings-per-share, to satisfy their basic need for affirmation and attention. However, Yook and Lee [43] showed that narcissistic CEOs affect financial performance measures through real activities such as operational changes rather than accrual-based manipulation.

More recently, researchers have begun investigating the influence of CEO narcissism on firm performance. Some scholars have pointed out that CEO narcissism can positively impact firm performance [44]. Narcissistic CEOs often enhance sustainability accounting and performance by engaging in real activities, such as promoting corporate social responsibility initiatives and adjusting sales management strategy [43]. Conversely, other scholars contend that CEO narcissism negatively influences firm performance [8]. Highly narcissistic CEOs tend to overestimate the value of the target and, hence, are willing to pay a higher M&A premium [45]. This behavior may result in suboptimal investments, ultimately diminishing the firm’s performance. Furthermore, these CEOs are more likely to pursue risky projects while neglecting core business operations, a tendency that can reduce investor confidence and result in lower firm valuations. Meanwhile, some scholars found that firm business performance is not influenced by the degree of CEOs’ narcissism [46]. Evidence shows that the relationship between CEO narcissism and firm value remains ambiguous. Additionally, scholars have begun to examine how CEO narcissism affects a firm’s R&D investment. Evidence suggests that narcissistic CEOs are particularly prone to invest in R&D expenditures [41]. Narcissistic CEOs desire to exhibit superiority and reaffirm their self-image, and risky bold moves offer a publicly visible route for pursuing R&D activities [42]. However, other researchers indicate a negative relationship between CEO narcissism and corporate R&D investment [47]. The narcissistic CEO’s ambition to rapidly expand the company and build a “business empire” often comes at the expense of R&D expenditures. These studies provide better evidence to understand the relationship between CEO narcissism, R&D investment, and firm performance. However, the existing literature has largely overlooked the mediating effect of R&D investment in the relationship between CEO narcissism and firm performance in Confucian cultural contexts, and neglected the moderating role of managerial discretion in the relationship between CEO narcissism and R&D investment. Therefore, this study aims to empirically examine three pivotal research questions. First, we investigate the influence of CEO narcissism on R&D investment and firm performance. Second, we analyze the mediating effect of R&D investment on the relationship between CEO narcissism and firm performance. Lastly, we evaluate the moderating role of managerial discretion in the link between CEO narcissism and R&D investment.

### 2.2. Effect of CEO Narcissism on Firm Performance

The upper echelons theory suggests that top managers, especially CEOs, operate under the constraints of bounded rationality, where their cognitions, values, and perceptions significantly shape their judgments and decisions, ultimately influencing firm performance [48]. A crucial component of CEO personality, especially narcissism, would affect firm performance. First, highly narcissistic CEOs often share their vision and publicize their personal experiences, which facilitates information exchange and enhances communication between the CEO and the top management team. This practice cultivates mutual understanding and maximizes resource utilization and integration, ultimately leading to improved firm performance. Second, highly narcissistic CEOs possess a heightened ability to identify and seize opportunities, as well as to anticipate and interpret market trends. This capability allows them to strengthen their firm’s competitive advantage in dynamic environments, ultimately enhancing overall performance [43]. Additionally, highly narcissistic CEOs often possess the ability to cultivate strong loyalty among their followers and garner support and cooperation from external stakeholders, resulting in a positive impact on firm performance. Finally, highly narcissistic CEOs tend to seek control and dominance, favoring leadership roles over those of followers. They are drawn to activities that attract significant social attention, aiming to garner praise and achieve high performance. Therefore, we proposed the following hypothesis.

**Hypothesis** **1.**
*CEO narcissism is positively associated with firm performance.*


### 2.3. Effect of CEO Narcissism on R&D Investment

R&D investment is fundamental to strategic decision-making, offering a unique opportunity for narcissistic CEOs to capture attention and admiration. First, narcissistic CEOs frequently prioritize boosting R&D investments as a means to attract attention and admiration. R&D investment is, by nature, a high-risk, high-reward endeavor that naturally garners extensive media coverage. This increased visibility not only helps these CEOs captivate public interest but also enables them to craft a charismatic image. Ultimately, this pursuit of recognition satisfies their personal aspirations for reputation, admiration, and acclaim. Second, narcissistic CEOs tend to engage in R&D activities to fulfill their need for dominance. Their inflated self-esteem compels them to dominate every facet of their lives. Specifically, narcissistic CEOs prefer to be actual leaders rather than followers. You et al. [43] highlight that R&D productivity can confer first-mover advantages and establish a firm’s market leadership. Finally, narcissistic CEOs may pursue R&D for self-serving purposes. Previous studies have demonstrated that investment in R&D enhances firm performance [49], which, in turn, can impact CEO compensation. Additionally, greater investment in R&D implies that narcissistic CEOs could widely mobilize organizational resources, which provides opportunities for self-interest. Therefore, we proposed the following hypothesis.

**Hypothesis** **2.**
*CEO narcissism is positively associated with R&D investment.*


### 2.4. Mediating Effect of R&D Investment

CEO narcissism, a significant personality trait among CEOs, can profoundly influence firm-level outcomes through organization activities [50]. For instance, Yook and Lee [43] propose that corporate social responsibility acts as a mediating factor between CEO narcissism and organizational performance. The accounting literature also shows that narcissistic CEOs are more likely to enhance financial performance metrics through real activities, such as stimulating sales growth [19]. This study proposes that R&D investment could be the missing link between CEO narcissism at the individual level and firm performance at the organizational level. We explain how R&D investment plays a mediating role in connecting narcissistic CEOs and firm performance. First, CEO narcissism may affect the willingness to engage in R&D investments, which may, in turn, enhance firm performance. CEOs with strong narcissistic traits often view themselves as unique or superior to their competitors, prompting them to make bold decisions that emphasize their exceptionalism. Engaging in R&D not only captures media and public attention but also serves as a platform for them to demonstrate their uniqueness, reinforcing their self-image and solidifying their perceived superiority. Second, CEO narcissism may shape the organization’s approach to R&D investments, ultimately impacting performance outcomes. Narcissistic CEOs have high levels of self-awareness, prefer adventures, and have foresight, which is conducive to supporting R&D projects. Moreover, highly narcissistic CEOs have significantly stronger tendencies to acquire and integrate R&D resources, ensuring sustained R&D activities and promoting growth in R&D investment and business performance. Therefore, we proposed the following hypothesis.

**Hypothesis** **3.**
*R&D investment will mediate the relationship between CEO narcissism and firm performance.*


### 2.5. Moderating Effect of Managerial Discretion

Managerial discretion plays a crucial role in strategic analysis and decision-making. High-discretion contexts increase the potential senior manager impact on strategic decision-making processes. The CEO, as the most powerful leader of the TMT, is particularly affected by managerial discretion. Finkelstein and Hambrick [51] contend that specific organizational attributes can either restrict or enhance a CEO’s managerial discretion at the organizational level. A firm’s governance structure largely reflects the organization’s discretion. Importantly, CEO duality and ownership are the crucial elements of the governance structure that can affect a CEO’s managerial discretion, which includes the relations between CEO narcissism and R&D.

CEO duality arises when the CEO simultaneously serves as the chairperson of the board, often leading to increased managerial discretion. We contend that CEO duality would enhance the positive influence of CEO narcissism on R&D. First, CEO duality enhances a narcissistic CEO’s power. CEO duality weakens the board’s monitoring role and the effectiveness of internal corporate governance and facilitates narcissistic CEOs to execute R&D strategies [52]. Moreover, CEO duality could significantly promote the narcissistic CEO’s risk appetite, and focus their attention on R&D investment to attract attention. Second, CEO duality can improve the narcissistic CEO’s effectiveness when making R&D investment decisions. Diversity within the top management team can impede communication, elongate the timeframe between R&D spending and innovative outcomes, and ultimately diminish R&D efficiency. CEO duality fosters concentrated attention on R&D investment-related decisions, and reduces inefficiencies in the decision-making process. Moreover, separating the roles of CEO and chairperson can introduce conflicts that detract from cohesive leadership. CEO duality, on the other hand, promotes leadership unity and consistency in decision-making, which ensures both the quality and rigor of R&D decisions.

CEO ownership plays a crucial role in mitigating agency problems, aligning the interests of the CEO with those of the shareholders, and preventing the expropriation of shareholder value. Elevated CEO ownership can enhance managerial discretion [53]. We argue that CEO ownership further heightens the positive relationship between CEO narcissism and R&D. Specifically, CEO ownership significantly impacts a firm’s risk appetite and R&D activities. CEOs pursue low-risk investments to maintain a stable salary when they do not hold the firm’s equity. In contrast, CEOs’ interests are aligned with company interests when CEOs hold shares in the companies. Narcissistic CEOs tend to lead their firms to pursue ‘high-risk, high-return’ investments, particularly in R&D projects, driven by the convergence effect. Furthermore, higher equity ownership amplifies the risk tolerance and investment capacity of narcissistic CEOs, leading to increased expenditures on R&D initiatives. Finally, narcissistic CEOs are often distinguished by their keen judgment, innovative mindset, and proactive approach to strategic decisions, enriching the R&D process. With the increase in the shareholding ratio, the control power held by narcissistic CEOs increases. Narcissistic CEOs with high shareholding can use their resource advantages to improve their competitive context, identify high-risk/high-return projects, and increase their R&D expenditure. Therefore, we proposed the following hypothesis.

**Hypothesis** **4a.**
*CEO duality positively moderates the relationship between CEO narcissism and R&D investment.*


**Hypothesis** **4b.**
*CEO ownership positively moderates the relationship between CEO narcissism and R&D investment.*


As mentioned above, we proposed a conceptual research framework (see Figure 1).

## 3. Methodology

### 3.1. Sample and Data

We selected A-share listed companies from 2011 to 2019 as our initial sample. The data collection process followed several exclusion criteria: (a) we excluded listed banks and insurance companies; (b) we removed companies labeled as ST and ST*; (c) we excluded firms that had been listed for less than three years, as well as CEOs who had served less than three years; (d) we eliminated firms with incomplete or abnormal data. The final sample is composed of 1243 firm-year observations. To alleviate the interference of extreme values, all continuous variables are winsorized at 1 percent and 99 percent. CEO narcissism data were collected using a video survey methodology, while additional data were obtained from the WIND and CSMAR databases. Data collection began on 25 May 2024, and concluded on 24 June 2024. During data collection, we recruited graduate students with experience in personality assessment to serve as raters. The recruitment process started on 27 May 2024, and ended on 31 May 2024.

### 3.2. Measures

#### 3.2.1. Firm Performance

Tobin’s Q (Tobin Q) reflects the average return on a company’s total capital and is often used to evaluate firm performance [1]. Drawing on previous empirical studies [54,55], we use Tobin’s Q as a proxy for evaluating firm performance. Tobin’s Q is calculated as the ratio of the market value of a company’s assets to the replacement cost of those assets. Higher Tobin’s Q ratios show superior long-run performance.

#### 3.2.2. CEO Narcissism

Most studies measuring narcissism use the Narcissistic Personality Inventory (NPI). However, in Chinese society, the term “narcissism” is often perceived negatively and as a sensitive topic, which may lead survey participants to guess the questionnaire’s intent and respond insincerely, resulting in skewed outcomes. Previous research in personality psychology has introduced short-form videos to showcase participants’ personalities to evaluators [56]. Recent evidence suggests that an individual’s narcissistic traits can be effectively gauged through their interactions with others. Meta-analytic research indicates that third-party ratings of personality traits exhibit significantly higher operational validity to self-reports [57]. Unlike self-reports, which can be inflated due to personal biases, third-party observers possess a more objective perspective, enabling them to identify targets’ personality traits. Third-party ratings of video samples can effectively be employed to measure CEO narcissism [40]. This novel video-metric approach allows us to overcome numerous limitations in measuring CEO narcissism [18]. This approach provides unobtrusive but direct access to large video samples of CEOs and accurate physical traces or documentary samples of CEOs [58]. Because video samples of CEOs are already widespread online, the availability and quality of CEO videos have been greatly improved by the widespread availability of video recording and editing tools. Therefore, it alleviates the reluctance of senior executives to participate in survey research. This approach also uses previously validated psychometric scales without concerns about a loss of responses based on sensitivity traits or social desirability response bias. Therefore, there is ample reason to believe that CEOs’ narcissistic tendencies can be assessed based on the video-metric approach. This novel video-metric approach has been successfully utilized in measuring CEO narcissism [18,40,50,58,59]. This approach requires trained raters to observe videos of CEOs of different lengths to give a numerical score on the items that measure CEO narcissism, which reflects perceived CEO narcissism by raters. Therefore, the video-metric approach presents a robust framework for quantifying the narcissistic tendencies of CEOs.

First, we identified a sample of 261 CEOs, compiling a total of 547 videos from public internet resources (the Baidu and Sougou search engines). Within one week, we edited these videos to remove any information that could bias the raters’ evaluations, such as the CEO’s name, company names, and job titles. The final dataset comprised 492 videos featuring 247 unique CEOs. These companies are mainly concentrated in economically developed regions in eastern China such as Guangdong Province, Shandong Province, Shanghai Municipality, and Zhejiang Province. These companies are clustered in the special equipment manufacturing industry, electrical machinery and equipment manufacturing, computer communications, and other electronic equipment manufacturing. Next, we employed the Narcissistic Personality Inventory (NPI-16), the prevailing instrument for measuring CEO narcissism [50]. To ensure the instrument’s relevance and accuracy, we conducted open-ended interviews with a group of scholars and senior managers with expertise in social psychology, allowing us to refine the 16-item questionnaire. This revised questionnaire was utilized by raters to evaluate the videos. The scale consists of four dimensions and includes 16 items. Items were scored on a 7-point Likert scale, where 1 = extremely disagree, 7 = strongly agree. These 16 items were broadly consistent with the four core dimensions of narcissism identified by Emmons [60]: entitlement/exploitativeness (i.e., they insist on getting respect), authority/leadership (i.e., they like to be the center of attention), superiority/arrogance (i.e., they are better than others), and self-admiration/self-absorption (i.e., they are preoccupied with how extraordinary and special they are). Table 1 presents Chinese CEOs’ 16-Item Narcissistic Personality Inventory (NPI-16) index.

Second, we recruited ten graduate students specializing in personality assessment to serve as raters. They were incentivized individually and divided into 5 groups. All raters underwent two days of training focused on evaluating each video based on the 16-item questionnaire. The training includes the following aspects: (1) the raters were asked to view the CEO video and assess four dimensions of narcissism, which included 16 items, using our revised Narcissistic Personality Inventory (NPI-16) in an Excel spreadsheet. The raters were informed of the meaning of the 16 items, and 16 items were used directly to assess CEO narcissism. The NPI-16 scale was developed in Excel to enable raters to input scores and provide specific justifications for each CEO’s rating. Specific justifications indicate why the rater assigned a particular score to the CEO. Examples of such justifications included whether a CEO emphasized their great contributions, used friendly facial expressions, exhibited straighter body posture, highlighted power and prestige, or wore colorful clothing [58]. (2) Raters were notified that this study focused on their perceptions of the CEOs in the video samples. There was no alternative interest in their responses and the raters were not evaluated. We also noted that this study was not a “trick”; rather, our main objective was to gather the raters’ perceptions. (3) We highlighted how to use a Likert-type response scale, ranging from 1 (strongly disagree) to 7 (strongly agree). Raters were instructed to use this scale to evaluate the extent to which they agreed with statements about the CEO in each video. (4) To familiarize raters with the task, we showed videos of three CEOs. We asked the raters to consider how to assess CEO narcissism using both the video materials and the narcissism scale. We also highlighted biases in rating (e.g., halo/horns effects; severity/leniency).

Third, we conducted pilot studies to determine acceptable video lengths and mitigate rater fatigue. Specifically, we categorized the CEO videos into 5 length groups: 1–3 min, 3–5 min, 5–10 min, 10–30 min, and over 30 min. A random selection of 30 videos was analyzed to analyze the differences across these lengths. The results indicated no significant variations in narcissism measures among the different videos lengths. Therefore, following Fung et al. [58], we evaluated each CEO’s videos for 5–10 min. Raters were provided with individual login codes to access the video samples using Qualtrics XM Platform. We randomly assigned CEOs to raters, who independently assessed the narcissistic tendencies of each CEO, completing all ratings within a two-week timeframe. If scores between the two raters exceeded the sample standard deviation, we reassessed the CEO’s narcissistic tendencies using two additional taters. If only one rater’s rating differed from the others, we deleted the rater’s score and averaged the ratings of the other three rater’s scores to derive the perceived narcissism index for that CEO. If there were significant differences among the four raters’ rating scores, we sourced additional video samples of the specific CEO from public sources and requested a reevaluate. Ultimately, the CEO’s narcissism score was calculated by averaging the scores across all four dimensions (16 items).

#### 3.2.3. R&D Investment

R&D plays a vital role in driving innovation and creating long-term value for companies. Following Yang et al. [61], R&D intensity (Rdi) was employed as the proxy for R&D investment. R&D intensity (Rdi) is defined as the ratio of a company’s R&D expenditures to its operating income. A higher R&D intensity indicates a stronger commitment to innovation.

#### 3.2.4. Managerial Discretion

The level of managerial autonomy conferred by CEO duality is examined by determining whether the CEO also holds the position of chairman. CEO duality (Dual) is represented as a dummy variable, taking a value of 1 if the CEO also serves as the chairperson of the board, and 0 otherwise. Furthermore, we measure the managerial autonomy associated with CEO ownership (Owner) as the percentage of outstanding shares owned by the CEO. Higher CEO ownership may effectively mitigate the pressure from concentrated shareholder interests and enhance their managerial autonomy.

#### 3.2.5. Control Variables

We introduced control variables that affect a firm’s performance mainly from four levels: firm, CEO, year and region. Control variables at the firm level include the firm’s size (Size), age (Fage), board size (Board), ownership concentration (Con), leverage (Lev), and profitability (Eps). Size is calculated as the natural logarithm of the firm’s operating revenue [62]. Fage represents the number of years since the establishment of the focal firm [58]. Board is measured as the natural logarithm of the number of board members [62,63]. Con is calculated by the proportion of shares held by the top ten shareholders relative to the total number of shares outstanding [64]. Lev is measured by the ratio of total liabilities to total assets [58]. Eps is measured by the actual earnings per share [65]. At the CEO level, three variables, the CEO’s gender (Gend), age (Age), and education (Edu), were chosen. Gend is a dummy variable, coded as 1 for males and 0 for females, and Age is the age of the CEO in years [58]. The Edu variable equals five if the CEO has a PhD degree, four for a master’s degree, three for a bachelor’s degree, two for an associate degree, and one otherwise. Finally, we introduced year (Year), and region (Region, at the province level) dummies [66].

### 3.3. Data Analysis

We utilized Stata 17.0 to estimate internal consistency (four dimensions of our 16-item questionnaire) and assess test–retest reliability. We employed Cronbach’s alpha as a measure of internal consistency; our sample yielded a Cronbach’s alpha of 0.8319, which exceeds the 0.70 threshold for acceptable reliability. Furthermore, we performed the Kaiser–Meyer–Olkin (KMO) test for sampling adequacy to evaluate the strength of the relationships among the items. The KMO statistic is 0.852, greater than 0.5, indicating that those 16 items have validity. Next, we matched the data on the CEO narcissism index, firm characteristics, and CEO-level variables. Stata 17.0 facilitated descriptive analyses, Pearson correlation assessments, hypothesis testing, and robustness checks.

## 4. Results

### 4.1. Descriptive Statistics and Correlation Analysis

Table 2 presents the correlations and descriptive statistics. The mean value of Tobin Q is 2.725, and the standard error is 2.144, indicating a diverse range of performances among publicly listed companies. The average score for CEO narcissism (Nar) is 4.450, suggesting that most CEOs in our sample exhibit relatively low levels of narcissism, with scores predominantly centered around the midpoint of 4 on a 7-point scale. Chinese society is rooted in Confucian philosophy and is largely collectivistic, so it is reasonable to expect minimal variation in narcissism scores among Chinese CEOs. Additionally, we measured CEO narcissism using five separate panels of raters. The first group’s scores range from a maximum of 5.25 to a minimum of 3.6875; the second group’s scores range from 5.5625 to 3.375; the third group’s scores range from 5.4375 to 3.5; the fourth group’s scores range from 5.4375 to 3.4375; and the fifth group’s scores range from 5.4375 to 3.625. These results indicate considerable variation in narcissism levels among CEOs. The mean value of Rdi is 0.074, suggesting that the sample firms exhibit relatively low levels of R&D investment. The mean Fage is 15.909, and the standard error is 5.394, highlighting significant variability in the ages of the firms. The mean Con is 63.970, indicating ownership concentration is significantly high across firms. The mean Lev is 36.020, reflecting the high leverage associated with Chinese firms. The means for Gend and Edu are 0.951, and 3.434, respectively, indicating that 95.10% of CEOs in our sample are male and that the CEOs generally possess high educational qualifications. No significant differences were observed among the other variables. Furthermore, the correlation coefficients between the variables are all below 0.50, suggesting weak multicollinearity. The mean VIF is 1.36, and the largest VIF is 2.43, further indicating the absence of substantial multicollinearity. Importantly, CEO narcissism is positively associated with firm performance, and statistically significant at the 1% level, which preliminarily verifies H1. However, CEO narcissism does not exhibit a significant correlation with Rdi, which will be examined further in subsequent analyses.

### 4.2. Regression Results

Table 3 presents the results for Hypotheses 1 to 4b. The dependent variable in Models 1–2 and 4–5 is Tobin Q, with Rdi for Models 3 and 6–7. We employed hierarchical regression analysis to assess the data. Step 1 is the baseline model, which included only control variables. We regressed six firm characteristics variables, namely firm size, firm age, board size, ownership concentration, firm leverage, and profitability, as well as CEO characteristics variables (i.e., CEO gender, CEO age, and CEO education), to the dependent variables. Step 2 included CEO narcissism as the independent variable. Then, R&D investment variables were included in Step 3. As illustrated in Model 1 of Table 3, three control variables, namely, Size (β = −0.348, *p* < 0.01), Lev (β = −0.031, *p* < 0.01), and Age (β = −0.028, *p* < 0.01), were found to be significantly and negatively associated with Tobin Q. Conversely, four control variables, namely, Con (β = 0.012, *p* < 0.01), Eps (β = 0.610, *p* < 0.01), Gend (β = 0.817, *p* < 0.01), and Edu (β = 0.198, *p* < 0.01), were found to be significantly and positively associated with Tobin Q. These findings align with the existing literature, which indicates that Size, Lev, and Age were found to negatively and significantly influence Tobin Q [67,68], while Con, Eps, Gend, Edu were found to positively and significantly influence Tobin Q [69,70,71]. Model 2 in Table 3 presents the results for Hypothesis 1, revealing that the regression equations possess 49.6% explanatory power and are statistically significant. CEO narcissism was positively associated with the Tobin Q rating at the 1% significance level. Our first hypothesis, which was that narcissistic CEOs could contribute to a firm’s performance, was confirmed. Hypothesis 2 predicted that CEO narcissism was positively related to R&D investment. Model 3 showed that the coefficient estimate of CEO narcissism was positive (β = 0.016, *p* < 0.01). Thus, our second hypothesis was confirmed. Additionally, Model 4 showed a positive correlation between Rdi and Tobin Q (β = 5.711, *p* < 0.01).

Following Baron and Kenny’s guidelines [72], we examined the mediating role of R&D investment in the relationship between CEO narcissism and firm performance. The following three conditions should be met to support mediation. First, CEO narcissism should significantly influence firm performance in the first-stage regression. Second, CEO narcissism should be associated with R&D investment in the second-stage regression. Third, CEO narcissism and R&D investment should influence firm performance simultaneously in the third-stage regression. Here, the independent variable of CEO narcissism was simultaneously entered as a predictor. Our analysis (Models 2 and 3 in Table 3) meet the first and second conditions of the Baron and Kenny criteria [72]. Importantly, when simultaneous regression was performed on Nar and Rdi (Model 5 in Table 3), the effect of Nar and Rdi remained positive and statistically significant (β = 0.521, *p* < 0.01; β = 5.384, *p* < 0.01), indicating that R&D mediated the effect of CEO narcissism on firm performance, thereby supporting our third hypothesis. Hypothesis 4a proposed that CEO duality positively moderates the relationship between CEO narcissism and R&D investment. We included CEO duality and the interaction variable (CEO narcissism × CEO duality) as independent variables. As shown in Model 6 of Table 3, we found the empirical support for H4a. More specifically, the interaction variable (Nar × Dual) was found to be positive and significant (β = 0.018, *p* < 0.1), thereby corroborating Hypothesis 4a. Hypothesis 4b suggested that CEO ownership positively moderates the impact of CEO narcissism on R&D investment. Again, in Model 7 of Table 3, the beta coefficient of the interaction effect of CEO narcissism and CEO ownership is positive and significant (β = 0.001, *p* < 0.01). Thus, Hypothesis 4b was supported.

### 4.3. Robustness Tests

To ensure the robustness of our findings, we conducted two analyses. First, we employed the substitution variable method to verify the reliability of our conclusions [73]. Following Chauvin and Hirschey [74], we re-operationalized Rdi. Rdi was calculated as the current year’s expenditures minus the previous year’s R&D expenditures, divided by total assets. The results, shown in Models 1–3 of Table 4, remained significant. Second, we utilized the lagged variable adjustment method to verify the robustness of our model. Following Schmidt and Pohler [75], we conducted lagged dependent variable and mediating variable regression analyses to estimate hypothesized effects. In this analysis, both R&D and Tobin Q were lagged by one year. The results, presented in Models 4–6 of Table 4, reveal substantial correlations with the earlier findings. Our empirical tests indicate that CEO narcissism is positively related to R&D and firm performance, and that R&D mediates the relationship between CEO narcissism and firm performance.

## 5. Discussion

### 5.1. Theoretical Contribution

Our research offers several contributions. First, our research contributes to the existing literature by demonstrating the significant influence of CEO narcissism on R&D investment. Previous studies have demonstrated the significance of various CEO characteristics (i.e., tenure, education, and experience) in influencing R&D investment [76,77,78,79], they have primarily relied on the traditional principal–agent theory. Consequently, they often overlook the concept of bounded rationality and fail to explore the influence of CEO personality traits. Drawing upon the upper echelon theory, top managers are individuals with bounded rationality, and their personality traits significantly influence the company’s strategic decisions and performance. Narcissism, a key component of executive personality traits, can significantly influence corporate strategic decision-making and performance. By examining the impact of CEO narcissism on R&D investments through the lens of the upper echelons theory, our study offers valuable insights into how CEO narcissism influences the behaviors of micro-enterprises. Furthermore, it expands upon the theoretical frameworks developed by Ham et al. [8] regarding the relationship between CEO narcissism and firms’ R&D investments. Second, this study enhances our understanding of the intrinsic relationship between CEO narcissism and firm performance. Previous studies have empirically confirmed the mediating roles of corporate social responsibility [40], entrepreneurial orientation [80], corporate learning strategies [81], and governance structures [82] in the relationship between CEO narcissism and corporate performance. However, they have overlooked the critical role of R&D investment in this dynamic. We have verified the mediating role of R&D investment in the relationship between CEO narcissism and firm performance, while also uncovering the value-creating effects of CEO narcissism on R&D investments. Additionally, it offers insights into the dynamics of CEO narcissism within emerging markets, aiding western CEOs in better comprehending their Chinese counterparts. Third, our study contributes to the fields of managerial discretion by highlighting the importance of CEO duality and CEO ownership in affecting the attitude and behavioral tendencies of narcissistic CEOs. We found that CEO duality and ownership amplify the effect of CEO narcissism on R&D investment. We contribute to a better understanding of the boundary conditions of CEO duality and CEO ownership on R&D investment in emerging economies. Thus, our study demonstrates that instead of examining the effect of CEO narcissism on risky decision-making behavior in isolation, it may be more insightful to explore the impact of CEO narcissism in conjunction with other factors that may either enhance or suppress such relationships.

### 5.2. Management Enlightenment

Our findings have meaningful implications for managers and policymakers. First, CEOs play a crucial role in decision-making, and CEO personality traits, especially narcissism, are imperative for the firm’s success in today’s highly innovative environment. Although CEOs with high narcissistic tendencies can probably adopt more aggressive innovation strategies, firms may encounter greater innovation investment risk and financing constraints. Narcissism is not necessarily a dark personality trait for CEOs, because R&D innovation activities require greater risks. Narcissistic CEOs are inclined to seek attention and praise through strategic investment (e.g., R&D), thus improving overall organizational performance. Therefore, when firms recruit a new CEO from outside, they should consider not only the CEO’s work experience and capabilities but also their psychological traits, particularly narcissism, which can influence the firm’s trajectory in R&D innovation and overall performance. Second, R&D investment has become a powerful tool for companies to improve performance. CEO decisions related to R&D must be based on external environmental dynamics and internal operation conditions. Narcissistic CEOs should enhance R&D risk assessment, enhance R&D cooperation, enhance R&D incentives, enhance R&D outcomes, and thus improve R&D performance. Further, narcissistic CEOs should overcome excessive narcissistic and ego-centered beliefs. More importantly, highly narcissistic CEOs should strengthen their self-evaluation through self-observation and self-prompts, thereby eliminating the impact of cognitive bias. Third, we found that CEO duality and CEO ownership grant narcissistic CEOs increased managerial discretion to implement the R&D investment activities they are interested in. However, a CEO’s managerial discretion in the Chinese context should be exercised reasonably. Corporations should strengthen CEOs’ managerial discretion when facing continual change from the business environment, and the specific context of breakthrough innovation. While the market regulatory mechanism remains immature and the company’s external governance mechanisms need improvement, the firm may deploy internal governance mechanisms to prevent the CEOs’ interests ahead of the company’s needs, restraining CEOs’ self-interest behavior. Fourth, both firm characteristics and CEO characteristics are key factors influencing firm performance. Therefore, decision-makers should thoughtfully develop strategies to enhance business performance and competitiveness, while dedicating attention and resources to activities that align with specific structures. In particular, our results suggest that ownership concentration, profitability, CEO gender, and CEO education are positively related to firm performance and merit special attention. Additionally, the company should focus uniquely on the “special” factors, such as firm size, firm leverage, and CEO age to eliminate their negative impact on firm performance.

## 6. Conclusions, Limitations, and Future Research

CEOs play a vital role in formulating and implementing firm’s strategies. Within the recent stream of work that focuses on the psychological traits, personal values, leadership style, and prior experiences of CEOs, narcissism has garnered significant attention as a pivotal and influential personality trait and can influence a CEO’s decision-making and actions. Narcissistic CEOs tend to exaggerate their abilities, prioritize self-serving objectives, crave attention, and motivated by self-interest and power. Prior research has mainly focused on the direct effects of CEO narcissism on R&D investments and form performance. Some studies suggest that CEO narcissism has a positive impact on R&D investments and firm performance [10,13], while other researchers have discovered that this same trait can result in a decrease in both R&D spending and performance [11,15]. These inconsistencies and contradictions may stem from differing cultural backgrounds and may overlook the impact of R&D investment and managerial discretion. Therefore, in this study, we employ the upper echelons theory to examine the impact of CEO narcissism on R&D investment and firm performance. Furthermore, we investigate the mediating role of R&D investment and the moderating effect of managerial discretion in this relationship. Our findings suggest that in the Chinese context, CEO narcissism positively affects both R&D investment and firm performance and that R&D investment can enhance enterprise performance. We suggested the role of investment as a core mediator in the relationship between CEO narcissism and firm performance. Narcissistic CEOs’ need for attention and image reinforcement may increase R&D investment and, as a result, enhance firm performance. We contributed to the R&D investment literature by integrating CEOs’ psychological personalities at the individual level and firm performance at the organizational level. Furthermore, this relationship between CEO narcissism and R&D investment might be even stronger under certain contingencies, such as the managerial discretion of CEOs granted by corporate governance factors. Thus, understanding the contingent model of CEO narcissism and its impact on R&D investment could be especially useful in helping companies with R&D and innovation in today’s competitive environment. In this context, we introduced CEO duality and ownership as a moderator of the relationship between CEO narcissism and R&D investment. We found that CEO duality and ownership positively moderate the relationship between CEO narcissism and R&D investment. Additionally, ownership concentration, profitability, CEO gender, and CEO education have a positive and significant managerial effect on firm performance, while some noteworthy factors such as firm size, firm leverage, and CEO age significantly and negatively affect firm performance.

Our study has several limitations. First, we have used the video-metric approach to measure CEO narcissism. This video-metric approach is gaining acceptance as a method for assessing CEO narcissism. This approach requires trained raters to watch video samples of CEOs and give a numerical score on the rating scale that measures narcissism, which reflects the raters’ perceptions. However, raters’ expertise levels and subjective attitudes may affect their assessment, resulting in bias when measuring CEO narcissism. Following Petrenko et al. [18], Gao et al. [40], Zhu and Chen [50], Fung et al. [58], and Gupta and Misangyi [59], we reduced rater bias in the following way. These strategies included utilizing suitable measurement scales, selecting professional raters, removing identifiable information from the videos, enhancing reliability by training, minimizing external interference, and thoroughly reviewing the rating data. However, some limitations remain. Future research should integrate the video-metric approach with non-intrusive methodologies (such as text analysis of speeches, letters and other communications) to analyze how narcissistic CEOs impact corporate strategic decision-making and performance. A second limitation was that we treated narcissism as a one-factor construct, as has also been carried out and validated by the extant research [20]. Future research might examine how different elements of the narcissism construct (such as entitlement/exploitativeness, authority/leadership, superiority/arrogance, and self-admiration/self-absorption) play different roles in affecting R&D investment and firm performance. Third, this paper has largely applied the upper echelons theory as a new approach. The literature, however, has explored R&D issues using a range of theories including resource dependence theory, agency theory, contingency theory, behavioral decision theory, and stakeholder theory. Future research may link CEOs’ personal traits to these well-developed theories to better understand the antecedents and consequences of R&D investment. Furthermore, previous studies have proven that a corporation’s top management team flourishing [83], stock liquidity [84], and strategic risk taking [85] also affect the firm’s performance; future studies should therefore include more mediation and control variables. Finally, the CEO and board chair are decision makers, but their job context, job attributes, and preferences are different, influencing their decision-making behavior. This paper examines the impact of CEO narcissism on R&D investment and firm performance. Future research should focus on selecting a sample of chairmen to investigate the influence of chairman narcissism on R&D and firm performance, and the mediating role of R&D in this relationship.

## Figures and Tables

**Figure 1 behavsci-14-01115-f001:**
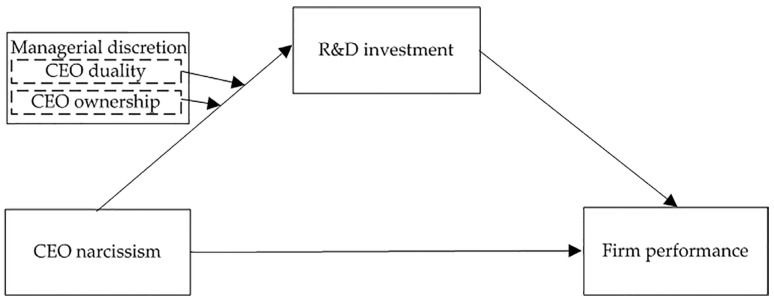
Research model.

**Table 1 behavsci-14-01115-t001:** Chinese CEOs’ 16-Item Narcissistic Personality Inventory (NPI-16) index.

Dimensions	Title Item
Entitlement/Exploitativeness	He/she usually dominates all conversations. People always seem to recognize his/her authority. He/she has a strong will to power. He/she likes to manipulate people.
Authority/Leadership	He/she has a high reputation in the industry. He/she likes people to obey him/her. He/she is more capable than others. He/she likes to establish authority over others.
Superiority/Arrogance	He/she likes to be followed. He/she is not willing to listen to subordinates. He/she is extraordinary and exceptional. He/she always knows what he/she is doing.
Self-admiration/Self-absorption	He/she always likes to be the center of attention. He/she sometimes gets angry when people complain about him/her. He/she is willing to express his/her opinion in public. He/she can usually be convinced to solve any problem.

**Table 2 behavsci-14-01115-t002:** Descriptive statistics and Pearson correlation matrix.

Variables	Mean	Sd	Tobin Q	Nar	Rdi	Dual	Owner	Size	Fage	Board	Con	Lev	Eps	Gend	Age	Edu
Tobin Q	2.725	2.144	1													
Nar	4.450	0.403	0.088 ***	1												
Rdi	0.074	0.070	0.272 ***	−0.013	1											
Dual	0.409	0.492	0.100 ***	0.031	0.023	1										
Owner	16.610	16.669	0.108 ***	0.068 **	−0.038	0.216 ***	1									
Size	21.515	1.473	−0.365 ***	0.059 **	−0.337 ***	−0.118 ***	−0.311 ***	1								
Fage	15.909	5.394	−0.227 ***	0.102 ***	−0.072 **	−0.04	−0.03	0.213 ***	1							
Board	2.121	0.184	−0.084 ***	−0.100 ***	−0.079 ***	−0.050 *	−0.113 ***	0.114 ***	0.043	1						
Con	63.970	14.468	0.098 ***	0.031	−0.263 ***	−0.017	0.053 *	0.124 ***	−0.245 ***	−0.025	1					
Lev	36.020	18.708	−0.459 ***	0.042	−0.280 ***	−0.114 ***	−0.219 ***	0.632 ***	0.298 ***	0.064 **	−0.105 ***	1				
Eps	0.515	0.626	0.139 ***	−0.016	−0.108 ***	0.051 *	−0.051 *	0.313 ***	−0.133 ***	0.133 ***	0.289 ***	−0.054 *	1			
Gend	0.951	0.216	−0.034	−0.055 *	−0.01	−0.001	−0.047 *	0.049 *	−0.02	0.042	−0.179 ***	0.079 ***	−0.029	1		
Age	52.101	6.883	−0.056 **	−0.009	−0.084 ***	−0.299 ***	−0.093 ***	0.044	0.155 ***	0.142 ***	0.042	−0.005	−0.007	0.012	1	
Edu	3.434	1.025	0.103 ***	−0.005	0.196 ***	0.150 ***	−0.063 **	0.056 **	−0.074 ***	−0.017	−0.121 ***	−0.041	0.055 *	0.100 ***	−0.211 ***	1

Notes: * *p* < 0.10, ** *p* < 0.05, *** *p* < 0.01.

**Table 3 behavsci-14-01115-t003:** Regression results.

Variables	Tobin Q	Tobin Q	Rdi	Tobin Q	Tobin Q	Rdi	Rdi
	Model 1	Model 2	Model 3	Model 4	Model 5	Model 6	Model 7
Size	−0.348 ***	−0.365 ***	−0.016 ***	−0.260 ***	−0.279 ***	−0.016 ***	−0.017 ***
	(−7.29)	(−7.71)	(−8.89)	(−5.59)	(−6.01)	(−9.12)	(−9.33)
Fage	−0.003	−0.011	−0.001 ***	0.004	−0.003	−0.001 ***	−0.001 ***
	(−0.26)	(−1.01)	(−4.04)	(0.36)	(−0.30)	(−3.69)	(−3.72)
Board	−0.124	0.032	−0.012	−0.031	0.098	−0.012	−0.018
	(−0.53)	(0.14)	(−1.04)	(−0.14)	(0.42)	(−1.02)	(−1.62)
Con	0.012 ***	0.012 ***	−0.001 ***	0.018 ***	0.018 ***	−0.001 ***	−0.001 ***
	(3.35)	(3.42)	(−5.97)	(4.99)	(4.99)	(−6.01)	(−6.19)
Lev	−0.031 ***	−0.031 ***	−0.000 ***	−0.028 ***	−0.028 ***	−0.000 ***	−0.001 ***
	(−8.87)	(−9.10)	(−3.34)	(−8.22)	(−8.42)	(−3.58)	(−4.34)
Eps	0.610 ***	0.626 ***	0.005 **	0.581 ***	0.596 ***	0.006 **	0.005 **
	(7.56)	(7.96)	(2.35)	(7.32)	(7.63)	(2.41)	(2.19)
Gend	0.817 ***	0.840 ***	−0.020 *	0.936 ***	0.949 ***	−0.021 *	−0.025 **
	(4.30)	(4.42)	(−1.78)	(4.90)	(5.00)	(−1.83)	(−2.39)
Age	−0.028 ***	−0.026 ***	0.000	−0.029 ***	−0.028 ***	−0.000	0.000
	(−3.84)	(−3.73)	(0.79)	(−4.05)	(−3.95)	(−0.06)	(0.65)
Edu	0.198 ***	0.190 ***	0.009 ***	0.143 ***	0.139 ***	0.010 ***	0.010 ***
	(4.46)	(4.30)	(5.39)	(3.41)	(3.33)	(5.73)	(5.79)
Nar		0.605 ***	0.016 ***		0.521 ***	0.011 **	0.011 **
		(5.29)	(3.62)		(4.64)	(2.05)	(2.21)
Rdi				5.711 ***	5.384 ***		
				(7.11)	(6.76)		
Dual						−0.013 ***	
						(−3.49)	
Nar × Dual						0.018 *	
						(1.95)	
Owner							−0.001 ***
							(−5.65)
Nar × Owner							0.001 ***
							(3.43)
_cons	10.948 ***	8.280 ***	0.409 ***	8.221 ***	6.079 ***	0.501 ***	0.536 ***
	(9.94)	(6.96)	(8.62)	(7.43)	(5.13)	(11.82)	(12.37)
Year/Region	Yes	Yes	Yes	Yes	Yes	Yes	Yes
N	1243	1243	1243	1243	1243	1243	1226
F	26.974	26.780	29.701	27.036	26.744	17.276	32.747
R^2^	0.503	0.514	0.313	0.527	0.535	0.323	0.342
Adj-R^2^	0.485	0.496	0.289	0.510	0.518	0.297	0.316

Note: * *p* < 0.10, ** *p* < 0.05, *** *p* < 0.01.

**Table 4 behavsci-14-01115-t004:** Robustness check.

Variables	Tobin Q	Rdi	Tobin Q	Tobin Q	Rdi	Tobin Q
	Model 1	Model 2	Model 3	Model 4	Model 5	Model 6
Size	−0.365 ***	−0.001	−0.433 ***	−0.396 ***	−0.000	−0.465 ***
	(−7.71)	(−1.40)	(−6.77)	(−6.64)	(−0.03)	(−6.65)
Fage	−0.011	−0.000	−0.010	−0.022 *	−0.000	−0.017
	(−1.01)	(−1.18)	(−0.70)	(−1.73)	(−1.08)	(−1.06)
Board	0.032	0.004	−0.073	0.034	0.004	0.005
	(0.14)	(1.50)	(−0.24)	(0.12)	(1.51)	(0.01)
Con	0.012 ***	−0.000	0.017 ***	0.010 **	−0.000	0.014 ***
	(3.42)	(−1.00)	(3.82)	(2.27)	(−0.81)	(2.85)
Lev	−0.031 ***	−0.000	−0.029 ***	−0.027 ***	−0.000	−0.024 ***
	(−9.10)	(−0.68)	(−6.55)	(−6.39)	(−1.18)	(−4.81)
Eps	0.626 ***	0.002 ***	0.612 ***	0.588 ***	0.001 *	0.575 ***
	(7.96)	(3.72)	(6.42)	(6.45)	(1.79)	(5.89)
Gend	0.840 ***	0.002	0.945 ***	0.808 ***	0.002	0.912 ***
	(4.42)	(1.10)	(4.04)	(3.57)	(1.04)	(3.70)
Age	−0.026 ***	−0.000	−0.018 **	−0.022 ***	−0.000	−0.017 *
	(−3.73)	(−0.25)	(−2.11)	(−2.72)	(−0.65)	(−1.85)
Edu	0.190 ***	0.001 *	0.178 ***	0.188 ***	0.001 **	0.162 ***
	(4.30)	(1.96)	(3.23)	(3.47)	(2.07)	(2.73)
Nar	0.605 ***	0.003 ***	0.682 ***	0.629 ***	0.003 ***	0.674 ***
	(5.29)	(3.48)	(4.23)	(4.73)	(3.33)	(3.78)
Rdi			39.370 ***			46.038 ***
			(4.11)			(4.37)
_cons	8.280 ***	−0.004	9.754 ***	9.184 ***	−0.013	10.737 ***
	(6.96)	(−0.45)	(6.56)	(6.64)	(−1.32)	(6.87)
Year/Region	Yes	Yes	Yes	Yes	Yes	Yes
N	1243	700	700	1018	657	657
F	26.780	5.560	20.267	22.600	4.182	18.358
R^2^	0.514	0.087	0.588	0.493	0.081	0.565
Adj-R^2^	0.496	0.039	0.565	0.472	0.029	0.540

Note: * *p* < 0.10, ** *p* < 0.05, *** *p* < 0.01.

## Data Availability

The data that support the findings of this study are available from the corresponding author (G.Z.) upon reasonable request.

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
