# Peer review of "Narcissistic Chief Executive Officers and Their Effects on R&D Investment and Firm Performance: The Moderating Role of Managerial Discretion"

_behavsci, 2024, doi:10.3390/bs14111115_

Round 1

Reviewer 1 Report (New Reviewer)

Comments and Suggestions for Authors

The paper studies the impact of narcissim on performance. Background is provided and explained well, as well as contribution of the study. The study uses a wide sample of CEOs and videos. The literature could showcace prior studies on personality, investment and performance. 

Was there anything found on other traits, or only narcissim? If not there are also leadership studies that have been done in relation to innovation and performance  that can be of use such as Entrepreneurial bricolage and entrepreneurial performance: The role of business model innovation and market orientation. Heliyon, and the paper How Transformational Leadership Affects Firm Innovation Performance - A Perspective Based on Environmental Dynamism and Business Model Innovation. Journal of Chinese Human Resources Management,

The methodology is a bit questionable with a lot of potential of bias. Do topics and themes of conversations from videos you collected cover content which the experts can relate to narcissim by observation. I assume that much content is completely irrelevant for infering anything, and that content analysis should have been done to eliminate parts which have nothing to do with the narcissistic traits. Or does ratio of general content in relation to narcissistic content play a role in their evluation?  Also some of the mentioned criteria of narcissim such as expertise and confidence are not neccessarily valid measures of this trait. Thus in this type of methodology thee is much potnetial for bias, and invalid findings.

You can make it more clear, and justify methodology by using past sources which started and justified this method and why, and also add to limitations. Currently limitations are too short to illustrate the real limitations of the paper.

Comments on the Quality of English Language

minor revision

Author Response

Comments 1: The paper studies the impact of narcissism on performance. Background is provided and explained well, as well as the contribution of the study. The study uses a wide sample of CEOs and videos. The literature could showcase prior studies on personality, investment, and performance.

Response 1: We agree with this comment. We deeply appreciate the reviewer’s suggestion. According to the reviewer’s comment, we have provided prior studies on personality, investment, and performance. (Lines 126-158, pages 3-4)

The upper-echelon theory suggests that CEO characteristics influence future firm performance through strategic actions [24]. Innovation is a key strategic activity for enterprises, catalyzing business growth, value creation, and establishing sustainable competitive advantages [25]. CEO characteristics are important factors affecting innovation and business performance. Prior studies have investigated the potential relationships between CEO personal characteristics, innovation behavior, and firm performance [26]. The distinctive traits of CEOs, such as their leadership styles [27], and personality traits [28], can significantly influence a firm’s innovation behavior and overall performance. Some scholars have found that the CEO’s leadership style can affect a company’s innovation and performance. For example, transformational leadership positively influences firm performance, and business model innovation as a mediator of this relationship [29]. Proactive CEOs can shape exploitative innovation and, to some extent, exploratory innovation, and its effect on firm performance is partially mediated by organizational ambidexterity [30]. Servant leadership positively impacts the performance of third-sector organizations, with innovative capacity as a mediating factor in this relationship [31]. Ethical leadership enhances environmental performance through green IT capital and technology innovation [32]. Entrepreneurs’ extraversion, conscientiousness, and openness positively affect innovativeness, and indirectly positively affect technological improvement and new technology adoption [33]. Some scholars have found that the personality traits of CEOs can significantly impact corporate innovation and performance. For example, overconfident CEOs can amplify the positive impact of the innovation premium on firm performance [34]. CEO humility promotes both exploitative and explorative innovation [35]. Executive hubris is positively correlated with firm innovation, however, this relationship diminishes in more complex environments [36]. Machiavellian CEOs can strengthen the relationship between strategic entrepreneurial behaviors and firm performance [37]. This literature deepens our understanding of the connections between CEO personal characteristics, innovative behavior, and firm performance. However, increasing scholars emphasize that studies should focus on the influence of CEOs’ dark personality traits. Shirokova et al. [37] emphasize that future research should concentrate on the CEOs’ dark traits and leadership because this could enhance our understanding of leadership’s role in complex and dynamic contexts. Narcissism is often considered an important component of dark personality traits. Generally, CEOs are more likely to exhibit narcissistic personality traits.

In the literature review section, we expanded existing research on CEO narcissism, R&D investment, and corporate performance. (Lines 188-219, pages 4-5)

More recently, researchers have begun investigating the influence of CEO narcissism on firm performance. Some scholars have pointed out that CEO narcissism can positively impact firm performance [44]. Narcissistic CEOs tend to improve sustainability accounting and performance through engagement in real activities, such as promoting corporate social responsibility initiatives and adjusting sales management strategy [43]. Other scholars firmly believe CEO narcissism negatively influences firm performance [8]. Highly narcissistic CEOs tend to overestimate the value of the target and, hence, are willing to pay a higher M&A premium [45]. This may lead to suboptimal investment and diminish the firm’s performance. Highly narcissistic CEOs are likely to favor risky projects while neglecting the core business, which can reduce investor confidence and reactions, resulting in lower firm value. Other scholars found that firm business performance is not influenced by the degree of CEOs’ narcissism [46]. Evidence shows that the relationship between CEO narcissism and firm value remains unclear. Additionally, scholars have explored the effect of CEO narcissism on a firm’s R&D investment. Researchers have found that narcissistic CEOs are particularly prone to invest in R&D expenditures [41]. Narcissistic CEOs desire to exhibit superiority and reaffirm their self-image, and risky bold moves offer a publicly visible route for pursuing R&D activities [42]. Researchers also indicated that CEO narcissism will be negatively related to corporate R&D investment [47]. The narcissistic CEO’s sense of superiority and vanity motivates them to aggressively pursue rapid company growth to build a “business empire”. However, this strategy of external expansion will reduce the firm's R&D spending. These studies provide better evidence to understand the relationship between CEO narcissism, R&D investment, and firm performance. However, existing literature largely overlooked the mediating effect of R&D investment behavior in the relationship between CEO narcissism and firm performance in Confucian cultural contexts, and neglected the moderating role of managerial discretion in the relationship between CEO narcissism and R&D investment behavior. Therefore, this study empirically examines three pivotal research questions. First, we explore the influence of CEO narcissism on R&D investment and firm performance. Second, we analyze the mediating effect of R&D investment on the relationship between CEO narcissism and firm performance. Lastly, we evaluate the moderating role of managerial discretion in the link between CEO narcissism and R&D investment.

Comments 2: Was there anything found on other traits, or only narcissism? If not there are also leadership studies that have been done in relation to innovation and performance that can be of use such as Entrepreneurial bricolage and entrepreneurial performance: The role of business model innovation and market orientation. Heliyon, and the paper How Transformational Leadership Affects Firm Innovation Performance - A Perspective Based on Environmental Dynamism and Business Model Innovation. Journal of Chinese Human Resources Management.

Response 2: We agree with this comment. We are extremely grateful to the reviewer for pointing out this problem. We undertook a comprehensive review of the existing literature, identifying several studies that explore the impact of CEO leadership styles and psychological traits on innovation and corporate performance, particularly within the realms of leadership and innovation. (Lines 109-142, page 3)

The upper-echelon theory suggests that CEO characteristics influence future firm performance through strategic actions [24]. Innovation is a key strategic activity for enterprises, catalyzing business growth, value creation, and establishing sustainable competitive advantages [25]. CEO characteristics are important factors affecting innovation and business performance. Prior studies have investigated the potential relationships between CEO personal characteristics, innovation behavior, and firm performance [26]. The distinctive traits of CEOs, such as their leadership styles [27], and personality traits [28], can significantly influence a firm’s innovation behavior and overall performance. Some scholars have found that the CEO’s leadership style can affect a company’s innovation and performance. For example, transformational leadership positively influences firm performance, and business model innovation as a mediator of this relationship [29]. Proactive CEOs can shape exploitative innovation and, to some extent, exploratory innovation, and its effect on firm performance is partially mediated by organizational ambidexterity [30]. Servant leadership positively impacts the performance of third-sector organizations, with innovative capacity as a mediating factor in this relationship [31]. Ethical leadership enhances environmental performance through green IT capital and technology innovation [32]. Entrepreneurs’ extraversion, conscientiousness, and openness positively affect innovativeness, and indirectly positively affect technological improvement and new technology adoption [33]. Some scholars have found that the personality traits of CEOs can significantly impact corporate innovation and performance. For example, overconfident CEOs can amplify the positive impact of the innovation premium on firm performance [34]. CEO humility promotes both exploitative and explorative innovation [35]. Executive hubris is positively correlated with firm innovation, however, this relationship diminishes in more complex environments [36]. Machiavellian CEOs can strengthen the relationship between strategic entrepreneurial behaviors and firm performance [37]. This literature deepens our understanding of the connections between CEO personal characteristics, innovative behavior, and firm performance. However, increasing scholars emphasize that studies should focus on the influence of CEOs’ dark personality traits. Shirokova et al. [37] emphasize that future research should concentrate on the CEOs’ dark traits and leadership because this could enhance our understanding of leadership’s role in complex and dynamic contexts. Narcissism is often considered an important component of dark personality traits. Generally, CEOs are more likely to exhibit narcissistic personality traits.

Meanwhile, we greatly appreciate the reference you provided, as it has proven to be highly beneficial to our research. As a result, we have included this citation in our work. 

Innovation is a key strategic activity for enterprises, catalyzing business growth, value creation, and establishing sustainable competitive advantages [25]. (Lines 127-129, page 3)

25. Wu, S.; Luo, Y.; Zhang, H.; Cheng, P. Entrepreneurial bricolage and entrepreneurial performance: The role of business model innovation and market orientation. Heliyon. 2024, 10, e26600. https://doi.org/10.1016/j.heliyon.2024.e26600. (Lines 763-764, page 17)

For example, transformational leadership positively influences firm performance, and business model innovation as a mediator of this relationship [29]. (Lines 135-136, page 3)

29. Dong, B. How Transformational Leadership Affects Firm Innovation Performance-A Perspective Based on Environmental Dynamism and Business Model Innovation. J. Chin. Hum. Resour. Ma. 2023, 14, 38-50. https://doi.org/10.47297/wspchrmWSP2040-800503.20231 402. (Lines 771-773, page 17)

Comments 3: The methodology is a bit questionable with a lot of potential of bias. Do topics and themes of conversations from videos you collected cover content which the experts can relate to narcissism by observation. I assume that much content is completely irrelevant for infering anything, and that content analysis should have been done to eliminate parts which have nothing to do with the narcissistic traits. Or does ratio of general content in relation to narcissistic content play a role in their evluation? Also some of the mentioned criteria of narcissim such as expertise and confidence are not neccessarily valid measures of this trait. Thus in this type of methodology thee is much potnetial for bias, and invalid findings. 

You can make it more clear, and justify methodology by using past sources which started and justified this method and why, and also add to limitations. Currently limitations are too short to illustrate the real limitations of the paper.

Response 3: We agree with this comment. Thank you for underlining this deficiency. We made the following modifications to strengthen the robustness of the method.

    First, we cite existing literature to demonstrate the validity of our approach. (Lines 356-379, page 8)

Previous research in personality psychology has employed short-form videos to showcase participants’ personalities to evaluators [56]. Recent evidence also suggests that an individual’s narcissistic personality can be observed through their interactions with others. Research utilizing meta-analysis has shown that third-party ratings of personality traits exhibit significantly higher operational validity to self-reports [57]. Unlike self-reports, which can be inflated due to personal biases, third-party observers possess a more objective perspective, enabling them to identify targets’ personality traits. Third-party ratings of video samples can effectively be employed to measure CEO narcissism [40]. This novel video-metric approach allows us to overcome numerous limitations in measuring CEO narcissism [18]. This approach provides unobtrusive but direct access to large video samples of CEOs and accurate physical traces or documentary samples of CEOs [58]. Because video samples of CEOs are already widespread online, the availability and quality of CEO videos have been greatly improved by the widespread availability of video recording and editing tools. Therefore, it alleviates the reluctance of senior executives to participate in survey research. This approach also uses previously validated psychometric scales without concerns about loss of responses based on sensitivity traits or social desirability response bias. Therefore, there is ample reason to believe that CEOs’ narcissistic tendencies can be assessed based on the video-metric approach. This novel video-metric approach has been used to measure CEO narcissism [18, 40, 50, 58, 59]. This approach requires trained raters to observe videos of CEOs of different lengths to give a numerical score on the items that measure CEO narcissism, which reflects perceived CEO narcissism by raters. Therefore, there is ample reason to believe that CEO’s narcissistic tendencies can be assessed based on the video-metric approach.

Second, we performed a content analysis to remove sections that were not relevant to narcissistic traits. (Lines 404-413, page 9)

Specifically, the training includes the following aspects: (1) the raters were asked to view the CEO video and assess four dimensions of narcissism, which included 16 items, using our revised Narcissistic Personality Inventory (NPI-16) in an Excel spreadsheet. The raters were informed of the meaning of the 16 items, and 16 items were used directly to assess CEO narcissism. The NPI-16 scale was developed in Excel to enable raters to input scores and provide specific justifications for each CEO’s rating. Specific justifications indicate why the rater assigned a particular score to the CEO. Examples of such justifications included whether a CEO emphasized their great contributions, used friendly facial expressions, exhibited straighter body posture, highlighted power and prestige, or wore colorful clothing [58].

Third, we enhance the development of our narcissism indicator to increase both the accuracy and validity of measuring CEO narcissism. (Lines 394-399, page 8)

These 16 items were broadly consistent with the four core dimensions of narcissism identified by Emmons [60]: entitlement/exploitativeness (i.e., he/she insists on getting respect due to him/herself), authority/leadership (i.e., he/she likes to be the center of attention), superiority/arrogance (i.e., he/she is better than others), and self-admiration/ self-absorption (i.e., he/she is preoccupied with how extraordinary and special).

Fourth, we optimized the research process to reduce potential bias and enhance methodological rigor. (Lines 413-423, page 9)

(2) raters were notified that this study focused on their perceptions of the CEOs in the video samples. There was no alternative interest in their responses and raters will not be evaluated. We also noted that this study was no “trick” but, our main objective was to gather the raters' perceptions. (3) we highlighted how to use a Likert-type response scale, ranging from 1 (extremely disagree) to 7 (strongly agree). Raters were instructed to use this scale to evaluate the extent to which they agreed with statements about the CEO in each video. (4) to familiarize raters with the task, we showed videos of three CEOs. We asked the raters to consider how to assess CEO narcissism using both the video materials and the narcissism scale. We also highlighted biases in rating (e.g., ha-lo/horns effects; severity/leniency).

Fifth, the videos we collected showcased content raters could identify as indicative of narcissistic behavior through their observations. On the one hand, narcissism is an important component of the CEO’s psychological traits that can reflect a CEO’s cognitions and values. CEOs are more likely to exhibit narcissistic personality traits. Therefore, CEO videos are linked to narcissism, though it's important to note that there are varying degrees of this trait. On the other hand, this study focused on raters' perceptions of the CEOs in the video samples. We also noted that this study was no “trick” but, our main objective was to gather the raters' perceptions. Therefore, I am particularly focused on the CEO’s level of narcissism.

Finally, we have made efforts to minimize potential bias; however, there are certain limitations associated with the article, which we address in the limitations section. (Lines 699-682, page 16)

First, we have used the video-metric approach to measure CEO narcissism. This vid-eo-metric approach is gaining acceptance as a method for assessing CEO narcissism. This approach requires trained raters to watch video samples of CEOs and give a numerical score on the rating scale that measures narcissism, which reflects the raters’ perceptions. However, raters’ expertise levels and subjective attitudes may affect their assessment, resulting in bias when measuring CEO narcissism. Following Petrenko et al. [18], Gao et al. [40], Zhu and Chen [50], Fung et al. [58], and Gupta and Misangyi [59], we reduced rater bias in the following way. These strategies included utilizing suitable measurement scales, selecting professional raters, removing identifiable information from the videos, enhancing reliability by training, minimizing external interference, and thoroughly reviewing the rating data. However, some limitations remain. Future research should integrate the video-metric approach with non-intrusive methodologies to analyze how narcissistic CEOs impact corporate strategic decision-making and organizational performance.

4. Response to Comments on the Quality of English Language

Point 1:

Response 1: We agree with this comment. We are extremely grateful to the reviewer for pointing out this problem. We reviewed the article for grammatical accuracy, refined the language, and enhanced the overall quality of the English.

For example:

Firm value is important in determining survival and growth. The upper echelons theory posits that CEO characteristics influence future firm performance through strategic actions [1]. (Lines 30-32, page 1)

Recent studies have shown that highly narcissistic CEOs think highly of their abilities, emphasize self-focused goals, desire to be the center of attention, and pursue self-interest and power. These tendencies will motivate narcissistic CEOs to engage in various activities, which may, in turn, influence firm performance [9,10]. (Lines 42-45, pages 1-2)

Another group of scholars found, (Lines 50, page 2)

Prior studies have tested the mediation effect of corporate social responsibility [18], earnings management [19], and ownership structure [6] in the relationship between CEO narcissism and firm performance, (Lines 72-74, page 2)

The impact of CEO narcissism on organizational outcomes has attracted the attention of scholars. (Lines 173-174, page 4)

R&D investment is a vital element of strategic decision-making. R&D investment offers a unique opportunity for narcissistic CEOs to garner attention and admiration. (Lines 240-241, page 5)

CEOs pursue low-risk investments to maintain a stable salary when they do not hold the firm’s equity. (Lines 311-312, page 7)

Table 3 presents the results for Hypothesis 1. (Lines 527, page 12)

this study enriches the existing literature by examining the diverse factors that shape firms’ decisions related to R&D investment. (Lines 589-590, page 14)

This study examines the influence of CEO narcissism on firm-level R&D investments through the upper echelons theory and bounded rationality assumption. It offers new insights into how CEO narcissism affects the behaviors of micro-enterprises. (Lines 593-596, page 14)

5. Additional clarifications

In addition to addressing the editor's and reviewers' comments, we have implemented several additional revisions.

First, we have enhanced the introduction by incorporating additional details regarding the research deficiencies, particularly highlighting the insufficient exploration of moderator variables. (Lines 80-86, page 2)

(3) Previous studies have neglected several key moderating factors that significantly influence the behavior of narcissistic CEOs. Narcissistic CEOs may face either resource constraints or governance constraints, these may affect the cognitive process of decision-making, which in turn affects R&D investment and firm performance. Cragun et al. [7] emphasize the importance of examining the factors that help narcissistic CEOs enforce their preference, underlining managerial discretion as an important factor.

Second, in the introduction, we explore in greater depth the contributions and innovations that are closely intertwined with the research topic. (Lines 109-117, page 3)

Our study enriches the research on the relationship between CEO narcissism and R&D and firm performance in the Chinese context. While the primary impacts of CEO narcissism on R&D investment and firm performance have garnered attention, much of the existing literature is mostly from developed country settings. Based on Confucian values and ideologies, we explore the impact of CEO narcissism on R&D investment and firm performance. Importantly, this study verifies that R&D investment mediates the positive relationship between CEO narcissism and firm performance. It provides insights into the “internal black box” of how CEO narcissism impacts firm performance and deepens our understanding of this intrinsic pathway.

Third, we have revised and updated Figure 1. (Lines 329-330, page 7)

Fourth, we updated our funding.

Liaoning Provincial Department of Education Graduate Research Innovation Special Project: Research on the Mechanisms of Impact and Improvement Strategies for Algorithm Control of Digital Labor Platforms on Gig Workers' Work Engagement (DUFEYJS24006). (Lines 696-698, page 16)

    Fifth, we updated the table numbers and enriched the references with new additions and revisions.

Reviewer 2 Report (New Reviewer)

Comments and Suggestions for Authors

Dear authors,

Thank you for the manuscript sent me to review. Below are some suggestions and comments for your consideration:

1. It is necessary to enrich the citations within the Introduction and Literature Review sections. This could focus on the relationships among the three main variables being tested.

2. The paragraph in lines 262-269 elaborate on the process of sample selection. Underlying principles or concepts are important in determining the selection criteria. the period during which the authors collected the data should be transparently disclosed.

3. Table 2 (Descriptive Statistics) does not need to be presented; it is sufficient to describe it within the text.

4. The conclusion should first explain the new findings that differ from previous studies results before addressing the limitations of the study in terms of methodology and outcomes.

Good luck!

Comments on the Quality of English Language

Proofreading may be required.

Author Response

Comments 1: It is necessary to enrich the citations within the Introduction and Literature Review sections. This could focus on the relationships among the three main variables being tested.

Response 1: We agree with this comment. Thank you for the suggestion. We have enriched the citations within the Introduction and Literature Review sections.

On the one hand, we enhanced the introduction section by incorporating citations that discuss the relationship among the three primary variables. (Lines 54-65, page 2)

Few scholars have observed that the impact of CEO narcissism on firm performance remains uncertain [12]. R&D investment is a critical strategic decision for organizations and can be significantly influenced by narcissistic CEOs. Several recent studies found that narcissistic CEOs crave attention and praise, tend to be dominant, and desire for status and power propels them to take risks and pursue R&D activities [13]. R&D investment presents a substantial opportunity to attract media attention and stakeholder reactions, which, in turn, would satisfy the CEO’s personal needs for attention, and praise [14]. Other studies have suggested a negative association between CEO narcissism and green innovation [15]. Narcissistic CEOs aggressively pursue rapid growth strategies to build a “business empire” [16]. However, this strategy of external expansion will reduce R&D spending. Recent studies also showed that CEO narcissism is not significantly related to R&D investment [17].

On the other hand, we have expanded the citations in the literature review section to encompass studies exploring the relationship among the three key variables. (Lines 188-219, pages 4-5)

More recently, researchers have begun investigating the influence of CEO narcissism on firm performance. Some scholars have pointed out that CEO narcissism can positively impact firm performance [44]. Narcissistic CEOs tend to improve sustainability accounting and performance through engagement in real activities, such as promoting corporate social responsibility initiatives and adjusting sales management strategy [43]. Other scholars firmly believe CEO narcissism negatively influences firm performance [8]. Highly narcissistic CEOs tend to overestimate the value of the target and, hence, are willing to pay a higher M&A premium [45]. This may lead to suboptimal investment and diminish the firm’s performance. Highly narcissistic CEOs are likely to favor risky projects while neglecting the core business, which can reduce investor confidence and reactions, resulting in lower firm value. Other scholars found that firm business performance is not influenced by the degree of CEOs’ narcissism [46]. Evidence shows that the relationship between CEO narcissism and firm value remains unclear. Additionally, scholars have explored the effect of CEO narcissism on a firm’s R&D investment. Researchers have found that narcissistic CEOs are particularly prone to invest in R&D expenditures [41]. Narcissistic CEOs desire to exhibit superiority and reaffirm their self-image, and risky bold moves offer a publicly visible route for pursuing R&D activities [42]. Researchers also indicated that CEO narcissism will be negatively related to corporate R&D investment [47]. The narcissistic CEO’s sense of superiority and vanity motivates them to aggressively pursue rapid company growth to build a “business empire”. However, this strategy of external expansion will reduce the firm's R&D spending. These studies provide better evidence to understand the relationship between CEO narcissism, R&D investment, and firm performance. However, existing literature largely overlooked the mediating effect of R&D investment behavior in the relationship between CEO narcissism and firm performance in Confucian cultural contexts, and neglected the moderating role of managerial discretion in the relationship between CEO narcissism and R&D investment behavior. Therefore, this study empirically examines three pivotal research questions. First, we explore the influence of CEO narcissism on R&D investment and firm performance. Second, we analyze the mediating effect of R&D investment on the relationship between CEO narcissism and firm performance. Lastly, we evaluate the moderating role of managerial discretion in the link between CEO narcissism and R&D investment

Comments 2: The paragraph in lines 262-269 elaborate on the process of sample selection. Underlying principles or concepts are important in determining the selection criteria. the period during which the authors collected the data should be transparently disclosed.

Response 2: We agree with this comment. We sincerely thank you for your dedicated efforts and invaluable suggestions, which have significantly improved the quality of this paper. As suggested, we have transparently disclosed the period during which the data was collected. (Lines 340-241, page 7)

Data collection began on May 25, 2024, and concluded on June 24, 2024. During data collection, we recruited graduate students with experience in personality assessment to serve as raters. The recruitment process started on May 27, 2024, and ended on May 31, 2024. 

Comments 3: Table 2 (Descriptive Statistics) does not need to be presented; it is sufficient to describe it within the text.

Response 3: We agree with this comment. Thank you for your suggestion. We have deleted Table 2 and integrated the means and standard deviations into Table 3 (Create New Table 2). Furthermore, we have revised this section’s content and reorganized the order of the tables throughout the text. (Lines 485-489, pages 10-11; Lines 509, page 11)

Modified content: Table 2 presents the correlations and descriptive statistics. The mean of Tobin Q is 2.725, and the standard error is 2.144, indicating that publicly listed companies have different performances. The average score for CEO narcissism (Nar) is 4.450, suggesting that most CEOs in our sample display low levels of narcissism, with their scores predominantly centered around the midpoint of 4 on a 7-point scale. (Lines 485-489, pages 10-11)

Meanwhile, Table 2 has been revised. (Lines 509, page 12)

Table 2. Descriptive Statistics and Pearson Correlation Matrix.

Variables

Mean

Sd

Tobin Q

Nar

Rdi

Dual

Owner

Size

Fage

Board

Con

Lev

Eps

Gend

Age

Edu

Tobin Q

2.725

2.144

1

Nar

4.450

0.403

0.088***

1

Rdi

0.074

0.070

0.272***

-0.013

1

Dual

0.409

0.492

0.100***

0.031

0.023

1

Owner

16.610

16.669

0.108***

0.068**

-0.038

0.216***

1

Size

21.515

1.473

-0.365***

0.059**

-0.337***

-0.118***

-0.311***

1

Fage

15.909

5.394

-0.227***

0.102***

-0.072**

-0.04

-0.03

0.213***

1

Board

2.121

0.184

-0.084***

-0.100***

-0.079***

-0.050*

-0.113***

0.114***

0.043

1

Con

63.970

14.468

0.098***

0.031

-0.263***

-0.017

0.053*

0.124***

-0.245***

-0.025

1

Lev

36.020

18.708

-0.459***

0.042

-0.280***

-0.114***

-0.219***

0.632***

0.298***

0.064**

-0.105***

1

Eps

0.515

0.626

0.139***

-0.016

-0.108***

0.051*

-0.051*

0.313***

-0.133***

0.133***

0.289***

-0.054*

1

Gend

0.951

0.216

-0.034

-0.055*

-0.01

-0.001

-0.047*

0.049*

-0.02

0.042

-0.179***

0.079***

-0.029

1

Age

52.101

6.883

-0.056**

-0.009

-0.084***

-0.299***

-0.093***

0.044

0.155***

0.142***

0.042

-0.005

-0.007

0.012

1

Edu

3.434

1.025

0.103***

-0.005

0.196***

0.150***

-0.063**

0.056**

-0.074***

-0.017

-0.121***

-0.041

0.055*

0.100***

-0.211***

1

    Additionally, the order and content of the table below have been updated. 

Comments 4: The conclusion should first explain the new findings that differ from previous studies results before addressing the limitations of the study in terms of methodology and outcomes.

Response 4: We agree with this comment. In the conclusion section, we underscore our novel findings that diverge from previous studies and critically evaluate the limitations in terms of both methodology and results.

Prior research has mainly focused on the direct effects of CEO narcissism on R&D investments and form performance. However, the findings have not been consistent. Moreover, few studies have investigated how CEO narcissism impacts firm performance indirectly through R&D investment, and no studies have examined the role of managerial discretion. (Lines 646-650, pages 15)

Our study has several limitations. First, we have used the video-metric approach to measure CEO narcissism. This video-metric approach is gaining acceptance as a method for assessing CEO narcissism. This approach requires trained raters to watch video samples of CEOs and give a numerical score on the rating scale that measures narcissism, which reflects the raters’ perceptions. However, raters’ expertise levels and subjective attitudes may affect their assessment, resulting in bias when measuring CEO narcissism. Following Petrenko et al. [18], Gao et al. [40], Zhu and Chen [50], Fung et al. [58], and Gupta and Misangyi [59], we reduced rater bias in the following way. These strategies included utilizing suitable measurement scales, selecting professional raters, removing identifiable information from the videos, enhancing reliability by training, minimizing external interference, and thoroughly reviewing the rating data. However, some limitations remain. Future research should integrate the video-metric approach with non-intrusive methodologies to analyze how narcissistic CEOs impact corporate strategic decision-making and organizational performance. Second, the CEO and board chair are decision-makers, but their job context, job attributes, and preferences are different, influencing their decision-making behavior. This paper examines the impact of CEO narcissism on R&D investment and firm performance. Future research should focus on selecting a sample of chairmen to investigate the influence of chairman narcissism on R&D and firm performance, and the mediating role of R&D in this relationship. (Lines 669-687, pages 16)

4. Response to Comments on the Quality of English Language

Point 1:

Response 1: we agree with this comment. We are extremely grateful to the reviewer for pointing out this problem. We reviewed the article for grammatical accuracy, refined the language, and enhanced the overall quality of the English.

For example:

Firm value is important in determining survival and growth. The upper echelons theory posits that CEO characteristics influence future firm performance through strategic actions [1]. (Lines 30-32, page 1)

Recent studies have shown that highly narcissistic CEOs think highly of their abilities, emphasize self-focused goals, desire to be the center of attention, and pursue self-interest and power. These tendencies will motivate narcissistic CEOs to engage in various activities, which may, in turn, influence firm performance [9,10]. (Lines 42-45, pages 1-2)

Another group of scholars found, (Lines 50, page 2)

Prior studies have tested the mediation effect of corporate social responsibility [18], earnings management [19], and ownership structure [6] in the relationship between CEO narcissism and firm performance, (Lines 72-74, page 2)

The impact of CEO narcissism on organizational outcomes has attracted the attention of scholars. (Lines 173-174, page 4)

R&D investment is a vital element of strategic decision-making. R&D investment offers a unique opportunity for narcissistic CEOs to garner attention and admiration. (Lines 240-241, page 5)

CEOs pursue low-risk investments to maintain a stable salary when they do not hold the firm’s equity. (Lines 311-312, page 7)

Table 3 presents the results for Hypothesis 1. (Lines 527, page 12)

this study enriches the existing literature by examining the diverse factors that shape firms’ decisions related to R&D investment. (Lines 589-590, page 14)

This study examines the influence of CEO narcissism on firm-level R&D investments through the upper echelons theory and bounded rationality assumption. It offers new insights into how CEO narcissism affects the behaviors of micro-enterprises. (Lines 593-596, page 14)

5. Additional clarifications

In addition to addressing the editor's and reviewers' comments, we have implemented several additional revisions.

First, we have enhanced the introduction by incorporating additional details regarding the research deficiencies, particularly highlighting the insufficient exploration of moderator variables. (Lines 80-86, page 2)

(3) Previous studies have neglected several key moderating factors that significantly influence the behavior of narcissistic CEOs. Narcissistic CEOs may face either resource constraints or governance constraints, these may affect the cognitive process of decision-making, which in turn affects R&D investment and firm performance. Cragun et al. [7] emphasize the importance of examining the factors that help narcissistic CEOs enforce their preference, underlining managerial discretion as an important factor.

Second, in the introduction, we explore in greater depth the contributions and innovations that are closely intertwined with the research topic. (Lines 109-117, page 3)

Our study enriches the research on the relationship between CEO narcissism and R&D and firm performance in the Chinese context. While the primary impacts of CEO narcissism on R&D investment and firm performance have garnered attention, much of the existing literature is mostly from developed country settings. Based on Confucian values and ideologies, we explore the impact of CEO narcissism on R&D investment and firm performance. Importantly, this study verifies that R&D investment mediates the positive relationship between CEO narcissism and firm performance. It provides insights into the “internal black box” of how CEO narcissism impacts firm performance and deepens our understanding of this intrinsic pathway.

Third, we have revised and updated Figure 1. (Lines 329-330, page 7)

Fourth, we updated our funding.

Liaoning Provincial Department of Education Graduate Research Innovation Special Project: Research on the Mechanisms of Impact and Improvement Strategies for Algorithm Control of Digital Labor Platforms on Gig Workers' Work Engagement (DUFEYJS24006). (Lines 696-698, page 16)

    Fifth, we updated the table numbers and enriched the references with new additions and revisions.

Round 2

Reviewer 1 Report (New Reviewer)

Comments and Suggestions for Authors

revision made

Comments on the Quality of English Language

no comment

Author Response

3. Point-by-point response to Comments and Suggestions for Authors

Comments 1: Are the arguments and discussion of findings coherent, balanced and compelling? (Can be improved)

Response 1: We agree with this comment. We have strengthened the discussion of the results in the Discussion section, including rewriting the section and adding references to highlight the research contributions. (Lines 581-601, page 14)

First, our research contributes to the existing literature by demonstrating the significant influence of CEO narcissism on R&D investment. Previous studies have demonstrated the significance of various CEO characteristics (i.e., tenure, education, and experience) in influencing R&D investment [76-79], they have primarily relied on traditional principal-agent theory. Consequently, they often overlook the concept of bounded rationality and fail to explore the influence of CEO personality traits. Draw-ing upon the upper echelon theory, top managers are individuals with bounded rationality, and their personality traits significantly influence the company’s strategic decisions and performance. Narcissism, a key component of executive personality traits, can significantly influence corporate strategic decision-making and performance. By examining the impact of CEO narcissism on R&D investments through the lens of upper echelons theory, our study offers valuable insights into how CEO narcissism in-fluences the behaviors of micro-enterprises. Furthermore, it expands upon the theoretical frameworks developed by Ham et al. [8] regarding the relationship between CEO narcissism and firms’ R&D investments. Second, this study enhances our understanding of the intrinsic relationship between CEO narcissism and firm performance. Previous studies have empirically confirmed the mediating roles of corporate social responsibility [40], entrepreneurial orientation [80], corporate learning strategies [81], and governance structures [82] in the relationship between CEO narcissism and corporate performance. However, they have overlooked the critical role of R&D investment in this dynamic.

Comments 2: Are the conclusions thoroughly supported by the results presented in the article or referenced in secondary literature? (Can be improved)

Response 2: We agree with this comment. We enhanced the presentation of the conclusions to highlight the research focus, strengthen the relationship between the conclusions and results, and outline the research direction.

    On the one hand, we deepen our research conclusions. CEOs play a vital role in formulating and implementing firm’s strategies. With in the recent stream of work that focuses on the psychological traits, personal values, leadership style, and prior experiences of CEOs, narcissism has garnered significant attention as a pivotal and influential personality trait and can influence a CEO’s decision-making and actions. Narcissistic CEOs tend to exaggerate their abilities, prioritize self-serving objectives, crave attention, and motivated by self-interest and power. Prior research has mainly focused on the direct effects of CEO narcissism on R&D investments and form performance. Some studies suggest that CEO narcissism has a positive impact on R&D investments and firm performance [10, 13], while other researchers have discovered that this same trait can result in a decrease in both R&D spending and performance [11, 15]. These inconsistencies and contradictions may stem from differing cultural backgrounds and may overlook the impact of R&D investment and managerial discretion. Therefore, in this study, we employ upper echelons theory to exam-ine the impact of CEO narcissism on R&D investment and firm performance. Further-more, we investigate the mediating role of R&D investment and the moderating effect of managerial discretion in this relationship. Our findings suggest that in the Chinese context, (Lines 653-668, pages 15-16)

On the other hand, we elaborate on the research direction more comprehensively. A second limitation was that we treated narcissism as a one-factor construct, as extant research has also done and validated [20]. Future research might examine how differ-ent elements of the narcissism construct (such as entitlement/exploitativenes, authori-ty/leadership, superiority/arrogance, and self-admiration/ self-absorption) play differ-ent roles in affecting R&D investment and firm performance. Third, this paper is largely applied upper-echelon theory as a new approach. The literature, however, has explored R&D issues using a range of theories including resource dependence theory, agency theory, contingency theory, behavioral decision theory, stakeholder theory. Future research may link CEOs’ personal traits to these well-developed theories to better understand the antecedents and consequences of R&D investment. Furthermore, previous studies have proven that a corporation’s top management team flourishing [83], stock liquidity [84], strategic risk taking [85] also affect the firm’s performance, future studies should therefore include more mediation and control variables. (Lines 701-714, page 16)

4. Response to Comments on the Quality of English Language

Point 1: Moderate editing of English language required.

Response 1: We agree with this comment. In response to the reviewers' feedback highlighting the need for improvements in the English language, we have meticulously refined the grammar, content, and overall structure of the text.

For example:

Firm value is essential for a company’s survival and growth. (Lines 30, page 1)

Moreover, the leadership styles and psychological traits of CEOs can profoundly in-fluence a firm’s performance. Psychological research indicates that CEOs do not solely adhere to rational decision-making models, rather, their choices are profoundly shaped by their psychological traits [5, 6]. Narcissism is an important component of a CEO’s psychological profile, reflecting their cognitions and values. Also, CEO narcissism is characterized by grandiosity, self-focus, and self-importance [7, 8]. Recent evidence suggests that highly narcissistic CEOs tend to exaggerate their abilities, prioritize self-serving objectives, crave attention, and motivated by self-interest and power. (Lines 35-43, pages 1-2)

However, empirical literature yields mixed results. (Lines 47, page 2)

Highly narcissistic CEOs often assess risks and opportunities with heightened opti-mism and assertiveness[9], which will result in firm-beneficial outcomes [10]. Con-versely, other scholars found that CEO narcissism is negatively associated with firm performance. Highly narcissistic CEOs frequently engage in overinvestment to attract public attention, which can lead to inefficiencies in investment, ultimately, compro-mising organizational performance [8, 11]. Some scholars also observed that the impact of CEO narcissism on firm performance remains uncertain [12]. Furthermore, the in-fluence of CEO narcissism on R&D investment has become a focal point of interest. (Lines 49-56, page 2)

However, other research presents a contrasting view, indicating a negative correlation between CEO narcissism and green innovation [15]. Narcissistic CEOs may prioritize aggressive growth strategies aimed at building a “business empire” [16], which can ul-timately lead to a decrease in R&D spending. Moreover, some recent findings suggest that there may be no significant relationship between CEO narcissism and R&D in-vestment [17].

These inconsistent and contradictory effects may stem from several factors: (1) numerous studies have utilized survey data from various cultural backgrounds, high-lighting how national cultural values can influence the personality traits of CEOs. For instance, narcissism is a trait often associated with Western cultures, whereas Chinese society is deeply rooted in Confucian principles and exhibits a largely collectivistic mindset. Consequently, CEOs from diverse cultural backgrounds possess distinct life experiences and values, which can lead to variations in their decision-making perfor-mance and work behaviors. (Lines 61-74, page 2)

Narcissistic CEOs may encounter resource constraints and governance limitations that influence their decision-making processes. (Lines 83-85, page 2)

Upper echelons theory posits that managerial discretion is a pivotal factor that determines the range of actions available to CEOs. (Lines 88-89, page 2)

We address this gap by examining the topic through the lens of Confucian values and ideologies, shedding light on how CEO narcissism influences R&D investment and firm performance. (Lines 113-116, page 3)

CEO characteristics emerge as determinants of both innovation and business perfor-mance. Previous studies have investigated the potential relationships between CEOs’ personal traits, innovative behaviors, and firm performance [26]. CEO characteristics, including their leadership styles [27], and personality traits [28], can profoundly sig-nificantly influence a firm’s innovation strategies and performance. (Lines 130-135, page 3)

Upper echelons theory suggests that top managers, especially CEOs, operate under the constraints of bounded rationality, where their cognitions, values, and perceptions significantly shape their judgments and decisions, ultimately influencing firm performance [48]. A crucial component of CEO personality, especially narcissism, would affect firm performance. First, highly narcissistic CEOs often share their vision and publicize their personal experiences, which facilitates information exchange and enhances communication between the CEO and the top management team. This practice cultivates mutual understanding and maximizes resource utilization and integration, ultimately leading to improved firm performance. Second, highly narcissistic CEOs possess a heightened ability to identify and seize opportunities, as well as to anticipate and interpret market trends. This capability allows them to strengthen their firm’s compet-itive advantage in dynamic environments, ultimately enhancing overall performance [43]. Additionally, highly narcissistic CEOs often possess the ability to cultivate strong loyalty among their followers and garner support and cooperation from external stakeholders, resulting in a positive impact on firm performance. (Lines 221-235, page 5)

First, narcissistic CEOs frequently prioritize boosting R&D investments as a means to attract attention and admiration. R&D investment is, by nature, a high-risk, high-reward endeavor that naturally garners extensive media coverage. This increased visibility not only helps these CEOs captivate public interest but also enables them to craft a charismatic image. Ultimately, this pursuit of recognition satisfies their person-al aspirations for reputation, admiration, and acclaim. Second, narcissistic CEOs tend to engage in R&D activities to fulfill their need for dominance. Their inflated self-esteem compels them to dominate every facet of their lives. Specifically, narcissistic CEOs prefer to be actual leaders rather than followers. You et al. [43] highlight that R&D productivity can confer first-mover advantages and establish a firm’s market leadership. Finally, narcissistic CEOs may pursue R&D for self-serving purposes. Previous studies have demonstrated that investment in R&D enhances firm performance [49], which, in turn, can impact CEO compensation. (Lines 242-254, page 5)

CEOs with strong narcissistic traits often view themselves as unique or superior to their competitors, prompting them to make bold decisions that emphasize their exception-alism. Engaging in R&D not only captures media and public attention but also serves as a platform for them to demonstrate their uniqueness, reinforcing their self-image and solidifying their perceived superiority. Second, CEO narcissism may shape the organization’s approach to R&D investments, ultimately impacting performance out-comes. (Lines 270-276, page 6)

CEO duality arises when the CEO simultaneously serves as the chairperson of the board, often leading to increased managerial discretion. (Lines 293-294, page 6)

CEO ownership plays a crucial role in mitigating agency problems, aligning the interests of the CEO with those of the shareholders, and preventing the expropriation of shareholder value. Elevated CEO ownership can enhance managerial discretion [53]. (Lines 309-311, page 7)

Furthermore, higher equity ownership amplifies the risk tolerance and investment ca-pacity of narcissistic CEOs, leading to increased expenditures on R&D initiatives. Fi-nally, narcissistic CEOs are often distinguished by their keen judgment, innovative mindset, and proactive approach to strategic decisions, enriching the R&D process. (Lines 318-322, page 7)

We selected A-share listed companies from 2011 to 2019 as our initial sample. The data collection process followed several exclusion criteria: (a) we excluded listed banks and insurance companies; (b) we removed companies labeled as ST and ST⁎; (c) we ex-cluded firms that had been listed for less than three years, (Lines 336-339, page 7)

However, in Chinese society, the term “narcissism” is often perceived negatively and as a sensitive topic, which may lead survey participants to guess the questionnaire’s intent and respond insincerely, resulting in skewed outcomes. (Lines 356-359, page 8)

Therefore, the video-metric approach presents a robust framework for quantifying the narcissistic tendencies of CEOs.

First, we identified a sample of 261 CEOs, compiling a total of 547 videos from public internet resources (Lines 381-384, page 8)

Third, we conducted pilot studies to determine acceptable video lengths and mit-igate rater fatigue. Specifically, we categorized the CEO videos into 5 length groups: 1–3 minutes, 3–5 minutes, 5–10 minutes, 10–30 minutes, and over 30 minutes. A random selection of 30 videos was analyzed to analyze differences across these lengths. The results indicated no significant variations in narcissism measures among the different videos lengths. (Lines 427-432, page 9)

The level of managerial autonomy conferred by CEO duality is examined by de-termining whether the CEO also holds the position of chairman. CEO duality (Dual) is represented as a dummy variable, taking a value of 1 if the CEO also serves as chair-person of the board, and 0 otherwise. Furthermore, we measure the managerial au-tonomy associated with CEO ownership (Owner) as the percentage of outstanding shares owned by the CEO. (Lines 452-457, page 10)

Additionally, we measure CEO narcissism using five separate panels of raters. The first group’s scores range from a maximum of 5.25 to a minimum of 3.6875; the second group’s scores range from 5.5625 to 3.375; the third group’s scores range from 5.4375 to 3.5; the fourth group’s scores range from 5.4375 to 3.4375; and the fifth group’s scores range from 5.4375 to 3.625. These results indicate considerable variation in narcissism levels among CEOs. The mean value of Rdi is 0.074, suggesting that the sample firms exhibit relatively low levels of R&D investment. The mean Fage is 15.909, the standard error is 5.394, highlighting significant variability in the ages of the firms. The mean Con is 63.970, indicating ownership concentration is significantly high across firms. The mean Lev is 36.020, reflecting the high leverage associated with Chinese firms. (Lines 493-503, page 11)

Additionally, the article includes some minor language changes that are not outlined here. For more details, please refer to the manuscript.

5. Additional clarifications

First, we removed the word “behavior” from the title in order to better highlight the impact of CEO narcissism on R&D investment and the role of CEO narcissistic traits. New title is “Narcissistic Chief Executive Officers and Their Effects on R&D Investment and Firm Performance: The Moderating Role of Managerial Discretion.”

Second, we added citations to enrich the article.

  1. Wales, W. J.; Patel, P. C.; Lumpkin, G. T. In pursuit of greatness: CEO narcissism, entrepreneurial orientation, and firm performance variance. J. Manage. Stud. 2013, 50, 1041-1069. https://doi.org/10.1111/joms.12034.
  2. Wu, W.; Wang, H.; Wang, X. Entrepreneur narcissism and new venture performance: A learning perspective. J. Bus. Res. 2022, 149, 901-915. https://doi.org/10.1016/j.jbusres.2022.06.001.
  3. Chatterjee, A.; Pollock, T. G. Master of puppets: How narcissistic CEOs construct their professional worlds. Acad. Manage. Rev. 2017, 42, 703-725. https://doi.org/10.5465/amr.2015.022

4.

  1. 8 Christensen-Salem, A.; Kinicki, A.; Perrmann-Graham, J.; Walumbwa, F. CEO performance management behaviors’ influ-ence on TMT flourishing, job attitudes, and firm performance. Hum. Relat. 2023, 76, 1966-1989. https://doi.org/10.1177/00187267221119767.
  2. 84. Riaz, S.; Ali, R.; Hussain, S.; Rehman, R. Chief executive officer attributes, stock’s liquidity, and firm's performance. Manag. Decis. Econ. 2023, 44, 3397-3408. https://doi.org/10.100

2/mde.3886.

  1. Burkhard, B.; Sirén, C.; Van-Essen, M.; Grichnik, D.; Shepherd, D. A. Nothing ventured, nothing gained: A meta-analysis of ceo overconfidence, strategic risk taking, and performance. J. Manage. 2023, 49, 2629-2666. https://doi.org/10.1177/01492063221110203.

Reviewer 2 Report (New Reviewer)

Comments and Suggestions for Authors

Dear authors,

The recommendation outlined in the conclusions section for the future research lack sufficient depth. It would be greatly enhanced by a more comprehensive elaboration on methodological recommendations and the conceptual directions for subsequent research, as informed by the finding of the current study.

Comments on the Quality of English Language

Moderate. Proofreading may be necessary.

Author Response

3. Point-by-point response to Comments and Suggestions for Authors

Comments 1: The recommendation outlined in the conclusions section for the future research lack sufficient depth. It would be greatly enhanced by a more comprehensive elaboration on methodological recommendations and the conceptual directions for subsequent research, as informed by the finding of the current study.

Response 1: We agree with this comment. We have enriched the content of future research and pointed out the future research direction. (Lines 698-714, pages 16)

Future research should integrate the video-metric approach with non-intrusive meth-odologies (such as text analysis of speeches, letters and other communications) to analyze how narcissistic CEOs impact corporate strategic decision-making and performance. A second limitation was that we treated narcissism as a one-factor construct, as extant research has also done and validated [20]. Future research might examine how different elements of the narcissism construct (such as entitlement/exploitativenes, authority/leadership, superiority/arrogance, and self-admiration/ self-absorption) play different roles in affecting R&D investment and firm performance. Third, this paper is largely applied upper-echelon theory as a new approach. The literature, however, has explored R&D issues using a range of theories including resource dependence theory, agency theory, contingency theory, behavioral decision theory, stakeholder theory. Future research may link CEOs’ personal traits to these well-developed theories to better understand the antecedents and consequences of R&D investment. Furthermore, previous studies have proven that a corporation’s top management team flourishing [83], stock liquidity [84], strategic risk taking [85] also affect the firm’s performance, future studies should therefore include more mediation and control variables.

Comments 2: Is the content succinctly described and contextualized with respect to previous and present theoretical background and empirical research (if applicable) on the topic? 

Response 2: We agree with this comment. In the introduction, we will begin by offering a succinct overview of the research background, followed by a detailed examination of the specific research topic. We will also identify existing gaps in the current literature and outline the contributions our study aims to make.

Firm value is essential for a company’s survival and growth. (Lines 30, page 1)

    Moreover, the leadership styles and psychological traits of CEOs can profoundly in-fluence a firm’s performance. Psychological research indicates that CEOs do not solely adhere to rational decision-making models, rather, their choices are profoundly shaped by their psychological traits [5, 6]. Narcissism is an important component of a CEO’s psychological profile, reflecting their cognitions and values. Also, CEO narcissism is characterized by grandiosity, self-focus, and self-importance [7, 8]. Recent evidence suggests that highly narcissistic CEOs tend to exaggerate their abilities, prioritize self-serving objectives, crave attention, and motivated by self-interest and power. (Lines 35-43, page 1)

However, empirical literature yields mixed results. Scholars found that CEO narcissism could significantly increase firm performance. Highly narcissistic CEOs often assess risks and opportunities with heightened optimism and assertiveness [9], which will result in firm-beneficial outcomes [10]. Conversely, other scholars found that CEO nar-cissism is negatively associated with firm performance. Highly narcissistic CEOs fre-quently engage in overinvestment to attract public attention, which can lead to ineffi-ciencies in investment, ultimately, compromising organizational performance [8, 11]. Some scholars also observed that the impact of CEO narcissism on firm performance remains uncertain [12]. Furthermore, the influence of CEO narcissism on R&D invest-ment has become a focal point of interest. (Lines 47-56, page 2)

However, other research presents a contrasting view, indicating a negative correlation between CEO narcissism and green innovation [15]. Narcissistic CEOs may prioritize aggressive growth strategies aimed at building a “business empire” [16], which can ul-timately lead to a decrease in R&D spending. Moreover, some recent findings suggest that there may be no significant relationship between CEO narcissism and R&D in-vestment [17].

These inconsistent and contradictory effects may stem from several factors: (1) numerous studies have utilized survey data from various cultural backgrounds, high-lighting how national cultural values can influence the personality traits of CEOs. For instance, narcissism is a trait often associated with Western cultures, whereas Chinese society is deeply rooted in Confucian principles and exhibits a largely collectivistic mindset. Consequently, CEOs from diverse cultural backgrounds possess distinct life experiences and values, which can lead to variations in their decision-making performance and work behaviors. (Lines 61-74, page 2)

Comments 3: Are the research design, questions, hypotheses and methods clearly stated? (Can be improved)

Response 3: We agree with this comment. We clearly stated the study design, questions, hypotheses, and methods in the revision.

Upper echelons theory suggests that top managers, especially CEOs, operate under the constraints of bounded rationality, where their cognitions, values, and perceptions significantly shape their judgments and decisions, ultimately influencing firm performance. (Lines 221-224, page 5)

Second, highly narcissistic CEOs possess a heightened ability to identify and seize opportunities, as well as to anticipate and interpret market trends. This capability al-lows them to strengthen their firm’s competitive advantage in dynamic environments, ultimately enhancing overall performance [43]. Additionally, highly narcissistic CEOs often possess the ability to cultivate strong loyalty among their followers and garner support and cooperation from external stakeholders, resulting in a positive impact on firm performance. (Lines 229-235, page 5)

First, narcissistic CEOs frequently prioritize boosting R&D investments as a means to attract attention and admiration. R&D investment is, by nature, a high-risk, high-reward endeavor that naturally garners extensive media coverage. This increased visibility not only helps these CEOs captivate public interest but also enables them to craft a charismatic image. (Lines 242-246, page 5)

Their inflated self-esteem compels them to dominate every facet of their lives. Specifi-cally, narcissistic CEOs prefer to be actual leaders rather than followers. You et al. [43] highlight that R&D productivity can confer first-mover advantages and establish a firm’s market leadership. Finally, narcissistic CEOs may pursue R&D for self-serving purposes. Previous studies have demonstrated that investment in R&D enhances firm performance [49], which, in turn, can impact CEO compensation. (Lines 249-254, page 5)

CEOs with strong narcissistic traits often view themselves as unique or superior to their competitors, prompting them to make bold decisions that emphasize their exception-alism. Engaging in R&D not only captures media and public attention but also serves as a platform for them to demonstrate their uniqueness, reinforcing their self-image and solidifying their perceived superiority. Second, CEO narcissism may shape the organization’s approach to R&D investments, ultimately impacting performance out-comes. (Lines 270-276, page 6)

Furthermore, higher equity ownership amplifies the risk tolerance and investment capacity of narcissistic CEOs, leading to increased expenditures on R&D initiatives. Finally, narcissistic CEOs are often distinguished by their keen judgment, innovative mindset, and proactive approach to strategic decisions, enriching the R&D process. (Lines 318-322, page 7)

We selected A-share listed companies from 2011 to 2019 as our initial sample. The data collection process followed several exclusion criteria: (a) we excluded listed banks and insurance companies; (b) we removed companies labeled as ST and ST⁎; (c) we ex-cluded firms that had been listed for less than three years, as well as CEOs who had served less than three years. (Lines 336-340, page 7)

However, in Chinese society, the term “narcissism” is often perceived negatively and as a sensitive topic, which may lead survey participants to guess the questionnaire’s intent and respond insincerely, resulting in skewed outcomes. (Lines 356-359, page 8)

Therefore, the video-metric approach presents a robust framework for quantifying the narcissistic tendencies of CEOs.

First, we identified a sample of 261 CEOs, compiling a total of 547 videos from public internet resources (Baidu and Sougou search engines). (Lines 381-384, page 8)

To ensure the instrument’s relevance and accuracy, we conducted open-ended inter-views with a group of scholars and senior managers with expertise in social psychology, allowing us to refine the 16-item questionnaire. This revised questionnaire was utilized by raters to evaluate the videos. (Lines 393-396, page 8)

Comments 4: Are the arguments and discussion of findings coherent, balanced and compelling? (Can be improved)

Response 4: We agree with this comment. We have enhanced the coherence and balance between the arguments and the exploration of the findings.

On the one hand, we deepen the discussion of the findings and the research contribution in our theoretical contribution.

First, our research contributes to the existing literature by demonstrating the significant influence of CEO narcissism on R&D investment. Previous studies have demonstrated the significance of various CEO characteristics (i.e., tenure, education, and experience) in influencing R&D investment [76-79], they have primarily relied on traditional principal-agent theory. Consequently, they often overlook the concept of bounded rationality and fail to explore the influence of CEO personality traits. Drawing upon the upper echelon theory, top managers are individuals with bounded rationality, and their personality traits significantly influence the company’s strategic decisions and performance. Narcissism, a key component of executive personality traits, can significantly influence corporate strategic decision-making and performance. By examining the impact of CEO narcissism on R&D investments through the lens of upper echelons theory, our study offers valuable insights into how CEO narcissism in-fluences the behaviors of micro-enterprises. Furthermore, it expands upon the theoretical frameworks developed by Ham et al. [8] regarding the relationship between CEO narcissism and firms’ R&D investments. Second, this study enhances our understand-ing of the intrinsic relationship between CEO narcissism and firm performance. Previous studies have empirically confirmed the mediating roles of corporate social responsibility [40], entrepreneurial orientation [80], corporate learning strategies [81], and governance structures [82] in the relationship between CEO narcissism and corporate performance. However, they have overlooked the critical role of R&D investment in this dynamic. (Lines 581-601, page 14)

On the other hand, we deepen the conclusions in the research conclusion part and strengthen the relationship between arguments and conclusions.

CEOs play a vital role in formulating and implementing firm’s strategies. With in the recent stream of work that focuses on the psychological traits, personal values, leadership style, and prior experiences of CEOs, narcissism has garnered significant attention as a pivotal and influential personality trait and can influence a CEO’s decision-making and actions. Narcissistic CEOs tend to exaggerate their abilities, prioritize self-serving objectives, crave attention, and motivated by self-interest and power. Prior research has mainly focused on the direct effects of CEO narcissism on R&D investments and form performance. Some studies suggest that CEO narcissism has a positive impact on R&D investments and firm performance [10, 13], while other researchers have discovered that this same trait can result in a decrease in both R&D spending and performance [11, 15]. These inconsistencies and contradictions may stem from differing cultural backgrounds and may overlook the impact of R&D investment and managerial discretion. Therefore, in this study, we employ upper echelons theory to examine the impact of CEO narcissism on R&D investment and firm performance. Further-more, we investigate the mediating role of R&D investment and the moderating effect of managerial discretion in this relationship. Our findings suggest that in the Chinese context, (Lines 653-668, pages 15-16)

Comments 5: For empirical research, are the results clearly presented? (Can be improved)

Response 5: We agree with this comment. We clearly describe the empirical results. 

Additionally, we measure CEO narcissism using five separate panels of raters. The first group’s scores range from a maximum of 5.25 to a minimum of 3.6875; the second group’s scores range from 5.5625 to 3.375; the third group’s scores range from 5.4375 to 3.5; the fourth group’s scores range from 5.4375 to 3.4375; and the fifth group’s scores range from 5.4375 to 3.625. These results indicate considerable variation in narcissism levels among CEOs. The mean value of Rdi is 0.074, suggesting that the sample firms exhibit relatively low levels of R&D investment. The mean Fage is 15.909, the standard error is 5.394, highlighting significant variability in the ages of the firms. The mean Con is 63.970, indicating ownership concentration is significantly high across firms. The mean Lev is 36.020, reflecting the high leverage associated with Chinese firms. The means for Gend and Edu are 0.951, and 3.434, respectively, indicating that 95.10% of CEOs in our sample are male and that the CEOs generally possess high educational qualifications. No significant differences were observed among the other variables. Furthermore, the correlation coefficients between the variables are all below 0.50, sug-gesting weak multicollinearity. The mean VIF is 1.36, with the largest VIF is 2.43, fur-ther indicating the absence of substantial multicollinearity. Importantly, CEO narcis-sism is positively associated with firm performance, and statistically significant at the 1% level, which preliminarily verifies H1. However, CEO narcissism does not exhibit a significant correlation with Rdi, which will be examined further in subsequent analyses. (Lines 493-512, pages 11)

These findings align with existing literature, which indicates that Size, Lev, and Age were found to negatively and significantly influence Tobin Q [67-68], while Con, Eps, Gend, Edu were found to positively and significantly influence Tobin Q [69-71]. Model 2 in Table 3 presents the results for Hypothesis 1, revealing that the regression equations possess 49.6% explanatory power and are statistically significant. (Lines 528-533, pages 12)

We conducted two analyses. First, we employed the substitution variable method to verify the reliability of our conclusions. (Lines 565-566, pages 13)

Comments 6: Is the article adequately referenced? (Can be improved)

Response 6: We agree with this comment. We strengthen citations to enrich the references. (Lines 909-922, pages 20)

80. Wales, W. J.; Patel, P. C.; Lumpkin, G. T. In pursuit of greatness: CEO narcissism, entrepreneurial orientation, and firm performance variance. J. Manage. Stud. 2013, 50, 1041-1069. https://doi.org/10.1111/joms.12034.

81. Wu, W.; Wang, H.; Wang, X. Entrepreneur narcissism and new venture performance: A learning perspective. J. Bus. Res. 2022, 149, 901-915. https://doi.org/10.1016/j.

jbusres.2022.06.001.

82. Chatterjee, A.; Pollock, T. G. Master of puppets: How narcissistic CEOs construct their professional worlds. Acad. Manage. Rev. 2017, 42, 703-725. https://doi.org/10.5465/

amr.2015.0224.

83. Christensen-Salem, A.; Kinicki, A.; Perrmann-Graham, J.; Walumbwa, F. CEO performance management behaviors’ influ-ence on TMT flourishing, job attitudes, and firm performance. Hum. Relat. 2023, 76, 1966-1989. https://doi.org/10.1177/00187267221119767.

84. Riaz, S.; Ali, R.; Hussain, S.; Rehman, R. Chief executive officer attributes, stock’s liquidity, and firm's performance. Manag. Decis. Econ. 2023, 44, 3397-3408. https://doi.org/

10.1002/mde.3886.

85. Burkhard, B.; Sirén, C.; Van-Essen, M.; Grichnik, D.; Shepherd, D. A. Nothing ventured, nothing gained: A meta-analysis of ceo overconfidence, strategic risk taking, and performance. J. Manage. 2023, 49, 2629-2666. https://doi.org/10.1177/01492063221110203.

4. Response to Comments on the Quality of English Language

Point 1: Moderate editing of English language required.

Response 1: we agree with this comment. We are extremely grateful to the reviewer for pointing out this problem. We reviewed the article for grammatical accuracy, refined the language, and enhanced the overall quality of the English.

For example:

The impact of the chief executive officer (CEO) narcissism on a firm’s performance has gained attention from the academic community. (Lines 11-12, page 1)

Highly narcissistic CEOs often assess risks and opportunities with heightened opti-mism and assertiveness [9], which will result in firm-beneficial outcomes [10]. Con-versely, other scholars found that CEO narcissism is negatively associated with firm performance. Highly narcissistic CEOs frequently engage in overinvestment to attract public attention, which can lead to inefficiencies in investment, ultimately, compro-mising organizational performance [8, 11]. Some scholars also observed that the impact of CEO narcissism on firm performance remains uncertain [12]. Furthermore, the in-fluence of CEO narcissism on R&D investment has become a focal point of interest. (Lines 49-56, page 2)

Upper echelons theory posits that managerial discretion is a pivotal factor that determines the range of actions available to CEOs. (Lines 88-89, page 2)

We address this gap by examining the topic through the lens of Confucian values and ideologies, shedding light on how CEO narcissism influences R&D investment and firm performance. (Lines 113-116, page 3)

Empirical results support this hypothesis, thereby clarifying and reinforcing the rela-tionship between CEO narcissism and R&D investment while enriching our under-standing of the contextual factors influencing R&D investment. (Lines 122-124, page 3)

CEOs’ personal traits, innovative behaviors, and firm performance [26]. CEO charac-teristics, including their leadership styles [27], and personality traits [28], can profoundly significantly influence a firm’s innovation strategies and performance. (Lines 132-135, page 3)

Narcissistic CEOs often prioritize themes of power and self-centered objectives, driven by a strong desire for attention and praise. (Lines 161-162, page 4)

First, narcissism is regarded as a stable personality trait, whereas overconfidence tends to be more context-dependent [39]. Second, overconfident individuals tend to exhibit a sense of superiority without continuously seeking feedback from external sources to affirm their ego. (Lines 164-167, page 4)

They are self-centered, confident in their abilities, and often perceive themselves as superior. Motivationally, narcissistic CEOs constantly seek affirmation, praise, and ap-plause from others to create and maintain a sufficient “narcissistic supply”. The influ-ence of CEO narcissism on organizational outcomes has attracted significant attention from scholars. This phenomenon has been particularly noted in the context of strategic decision-making processes. (Lines 171-176, page 4)

However, other researchers indicate a negative relationship between CEO narcissism and corporate R&D investment [47]. The narcissistic CEO’s ambition to rapidly expand the company and build a “business empire” often comes at the expense of R&D expenditures. (Lines 206-209, page 5)

Upper echelons theory suggests that top managers, especially CEOs, operate under the constraints of bounded rationality, where their cognitions, values, and perceptions significantly shape their judgments and decisions, ultimately influencing firm performance [48]. A crucial component of CEO personality, especially narcissism, would affect firm performance. First, highly narcissistic CEOs often share their vision and publicize their personal experiences, which facilitates information exchange and enhances communication between the CEO and the top management team. This practice cultivates mutual understanding and maximizes resource utilization and integration, ultimately leading to improved firm performance. Second, highly narcissistic CEOs possess a heightened ability to identify and seize opportunities, as well as to anticipate and interpret market trends. This capability allows them to strengthen their firm’s competitive advantage in dynamic environments, ultimately enhancing overall performance [43]. Additionally, highly narcissistic CEOs often possess the ability to cultivate strong loyalty among their followers and garner support and cooperation from external stakeholders, resulting in a positive impact on firm performance. (Lines 221-235, page 5)

First, narcissistic CEOs frequently prioritize boosting R&D investments as a means to attract attention and admiration. R&D investment is, by nature, a high-risk, high-reward endeavor that naturally garners extensive media coverage. This increased visibility not only helps these CEOs captivate public interest but also enables them to craft a charismatic image. Ultimately, this pursuit of recognition satisfies their person-al aspirations for reputation, admiration, and acclaim. Second, narcissistic CEOs tend to engage in R&D activities to fulfill their need for dominance. Their inflated self-esteem compels them to dominate every facet of their lives. Specifically, narcissistic CEOs prefer to be actual leaders rather than followers. You et al. [43] highlight that R&D productivity can confer first-mover advantages and establish a firm’s market leadership. Finally, narcissistic CEOs may pursue R&D for self-serving purposes. Previous studies have demonstrated that investment in R&D enhances firm performance [49], which, in turn, can impact CEO compensation. (Lines 242-254, page 5)

CEOs with strong narcissistic traits often view themselves as unique or superior to their competitors, prompting them to make bold decisions that emphasize their exception-alism. Engaging in R&D not only captures media and public attention but also serves as a platform for them to demonstrate their uniqueness, reinforcing their self-image and solidifying their perceived superiority. Second, CEO narcissism may shape the organization’s approach to R&D investments, ultimately impacting performance out-comes. (Lines 270-276, page 6)

CEO duality arises when the CEO simultaneously serves as the chairperson of the board, often leading to increased managerial discretion. (Lines 293-294, page 6)

Diversity within the top management team can impede communication, elongate the timeframe between R&D spending and innovative outcomes, and ultimately diminish R&D efficiency. CEO duality fosters concentrated attention on R&D investment-related decisions, and reduces inefficiencies in the decision-making process. Moreover, separating the roles of CEO and Chairperson can introduce conflicts that detract from cohesive leadership. (Lines 301-306, page 6)

CEO ownership plays a crucial role in mitigating agency problems, aligning the interests of the CEO with those of the shareholders, and preventing the expropriation of shareholder value. Elevated CEO ownership can enhance managerial discretion [53]. (Lines 309-311, page 7)

Furthermore, higher equity ownership amplifies the risk tolerance and investment ca-pacity of narcissistic CEOs, leading to increased expenditures on R&D initiatives. Fi-nally, narcissistic CEOs are often distinguished by their keen judgment, innovative mindset, and proactive approach to strategic decisions, enriching the R&D process. (Lines 318-322, page 7)

We selected A-share listed companies from 2011 to 2019 as our initial sample. The data collection process followed several exclusion criteria: (a) we excluded listed banks and insurance companies; (b) we removed companies labeled as ST and ST⁎; (c) we ex-cluded firms that had been listed for less than three years, (Lines 336-339, page 7)

However, in Chinese society, the term “narcissism” is often perceived negatively and as a sensitive topic, which may lead survey participants to guess the questionnaire’s intent and respond insincerely, resulting in skewed outcomes. (Lines 357-359, page 8)

Therefore, the video-metric approach presents a robust framework for quantifying the narcissistic tendencies of CEOs.

First, we identified a sample of 261 CEOs, compiling a total of 547 videos from public internet resources. (Lines 381-384, page 8)

They were incentivized individually and divided into 5 groups. All raters underwent two days of training focused on evaluating each video based on the 16-item questionnaire. (Lines 406-408, page 9)

Third, we conducted pilot studies to determine acceptable video lengths and mit-igate rater fatigue. Specifically, we categorized the CEO videos into 5 length groups: 1–3 minutes, 3–5 minutes, 5–10 minutes, 10–30 minutes, and over 30 minutes. A random selection of 30 videos was analyzed to analyze differences across these lengths. The results indicated no significant variations in narcissism measures among the different videos lengths. (Lines 427-432, page 9)

The level of managerial autonomy conferred by CEO duality is examined by de-termining whether the CEO also holds the position of chairman. CEO duality (Dual) is represented as a dummy variable, taking a value of 1 if the CEO also serves as chair-person of the board, and 0 otherwise. Furthermore, we measure the managerial au-tonomy associated with CEO ownership (Owner) as the percentage of outstanding shares owned by the CEO. (Lines 452-457, page 10)

To ensure the robustness of our findings, we conducted two analyses. First, we employed the substitution variable method to verify the reliability of our conclusions [73]. (Lines 565-566, page 13)

Additionally, the article includes some minor language changes that are not outlined here. For more details, please refer to the manuscript.

5. Additional clarifications

    We removed the word “behavior” from the title in order to better highlight the impact of CEO narcissism on R&D investment and the role of CEO narcissistic traits. New title is “Narcissistic Chief Executive Officers and Their Effects on R&D Investment and Firm Performance: The Moderating Role of Managerial Discretion.”

This manuscript is a resubmission of an earlier submission. The following is a list of the peer review reports and author responses from that submission.

Round 1

Reviewer 1 Report

Comments and Suggestions for Authors

Thank you for your effort in conducting this interesting research, the literature review chapter is short but seems sufficient. However, some major revisions are recommended/needed to shed light on the hard work being done executing the data collection and analysis. It is also necessary for better understanding the procedure and how some computations being done.

The title: I suggest revisiting the title by consider replacing the ‘and’ with ‘on’, and make whatever needed adjustments accordingly

Chapter 3 Method: the methodology section rises various concerns:

1.     In the sampling section, please consider mentioning the total number of firms (which is mentioned in the abstract).

2.     The authors should provide the list of videos (the number of videos considered is also missing) along with the duration after editing and other descriptive data (industry if available, HQ region, etc. ) . Moreover, the min and max scores provided by students/evaluators in each group for each video could be added. 

3.     The authors need to mention / clarify if other studies has used the same video-metric approach. In addition, they need to add references or an acceptable reasoning on measuring R&D investment.  

4.     It seems that the authors used a questionnaire to collect data, around line # 295 you mentioned the dimensions and their description, it is a very good practice to list these dimensions in a tabular form along with their acronyms. It is also a good practice to share your edited version of the items in an Appendix even if it is in Chinese language, it will help other Chinese researchers. However, you must clarify what kind of edits were carried out on the items ( eg.; major/minor) other than removing any variables that could identify the video to evaluators.

5.     Line # 348, it is necessary to share the reliability scores and internal consistency for each dimension obtained from Stata maybe split Table – 1 in to two tables and add more descriptive statistics (most importantly max and min) along the mean and standard errors.   

6.     It is not clear if the control variable ‘firm age’ is computed in negative integers or decimals in years/months, the authors definition should be revisited and clarified, the authors definition of firm age is given as (Fage = firm establishment - the year of observation)?.

7.     Kindly, cite the method followed for robustness analysis. Moreover, I suggest that the author re-check the way this section is written, for example, the authors say, “another mediator”, maybe they mean the R&D variable is re-operationalized / re-calculated differently as…

8.     In general, the calculations of all variables should be defined with the currency or units of components involved in the computation.

Chapter 4 Results: The results section needs some improvement

1.     Line # 433 the authors mentions, “the above-mentioned studies”, there were no cited studies to refer to in the content preceding to this statement

2.     The analysis shows a low firm performance mean, Tobin Q is 2.725, with listing max and min in Table 1, a discussion of key findings should include whether the ‘low’ firm performance – if applies - is affected by other variables ‘low’ scores. This low-low effect also lead a positive coefficient, and should be explained as is, and thus, the managerial implications and conclusion chapter should be written with this in mind. 

Comments on the Quality of English Language

Sufficient but requires proof-reading. 
